# 3D Diffuser Actor: Multi-task 3D Robot Manipulation with Iterative Error Feedback

## Abstract

We present 3D Diffuser Actor, a framework that marries diffusion policies and 3D scene representations for robot manipulation. Diffusion policies capture the action distribution conditioned on the robot and environment state using conditional diffusion models. They have recently shown to outperform both deterministic and alternative generative policy formulations in learning from demonstrations. 3D robot policies use 3D scene feature representations aggregated from single or multiple 2D image views using sensed depth. They typically generalize better than their 2D counterparts in novel viewpoints. We unify these two lines of work and present a neural policy architecture that uses 3D scene representations to iteratively denoise robot 3D rotations and translations for a given language task description. At each denoising iteration, our model "grounds" the current end-effector estimate in the 3D scene workspace, featurizes it using 3D relative position attentions and predicts its 3D translation and rotation error. We test 3D Diffuser Actor on learning from demonstrations in simulation and in the real world. We show our model outperforms both 3D policies and 2D diffusion policies and sets a new state of the art on RLBench, an established learning from demonstrations benchmark, where it outperforms the previous SOTA with a 12% absolute gain. We ablate our architectural design choices, such as translation invariance through 3D grounding and relative 3D transformers, and show they help model generalization. Our results suggest that 3D scene representations and powerful generative modeling are key to efficient learning of multi-task robot policies.

## 1 Introduction

Many robot manipulation tasks are inherently multimodal: at any point during task execution, there may be multiple actions which yield task-optimal behavior. Demonstrations often contain diverse ways that a task can be accomplished. Learning from such multimodal data has been a persistent problem in imitation learning (Ho & Ermon, 2016; Tsurumine & Matsubara, 2022; Hausman et al., 2017; Shafiullah et al., 2022). A natural choice is then to treat policy learning as a distribution learning problem: instead of representing a policy as a deterministic map $\pi_\theta(x)$, learn instead the entire distribution of actions conditioned on the current robot state $p(y|x)$.

Diffusion models (Sohl-Dickstein et al., 2015; Ho et al., 2020) are a powerful class of generative models that learn to map noise to data samples from the distribution of interest, such as images (Dhariwal & Nichol, 2021; Ramesh et al., 2021; Rombach et al., 2022), videos (Ho et al., 2022; Singer et al., 2022), text (Lin et al., 2023) through iterative applications of a denoising function that takes as input the current noisy estimate and any relevant input conditioning information. Recent works that use diffusion models for learning robot manipulation policies from demonstrations (Pearce et al., 2023; Chi et al., 2023; Reuss et al., 2023) outperform their deterministic and generative alternatives, such as variational autoencoders (Mandlekar et al., 2019), mixture of Gaussians (Chernova & Veloso, 2007), combination of classification and regression objectives (Shafiullah et al., 2022), or energy based objectives (Florence et al., 2021). Specifically, they exhibit better distrubution coverage and higher fidelity than alternative formulations. They have so far been used with low-dimensional engineered state representations (Pearce et al., 2023) or 2D image encodings (Chi et al., 2023).

3D perception architectures that differentially map features from 2D perspective views to a 3D or bird's eye view (BEV) feature map are the de facto choice currently in autonomous driving perception

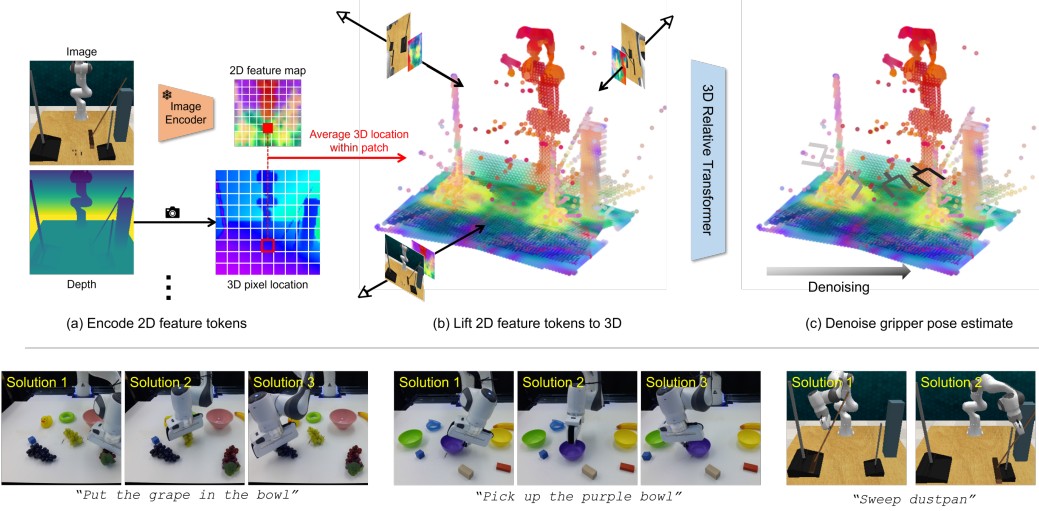

Figure 1: **3D Diffuser Actor** is a language and vision conditioned action diffusion model that learns multi-task multimodal end-effector keyposes from demonstrations. (a) We deploy an image encoder to extract a feature map from each camera view, where each feature token corresponds to a fixed-size patch in the input image. We compute the 3D locations of pixels by combining the 2D pixel locations and depth values using the camera intrinsics and pinhole camera equation. (b) We lift a 2D feature token to 3D by averaging the 3D locations of all pixels within the patch. (c) We apply a 3D relative transformer to denoise the robot gripper pose for each keyframe.

(Philion & Fidler, 2020; Li et al., 2022; Harley et al., 2023). Lifting features from perspective views to a BEV or 3D robot workspace map has also shown strong results in robot manipulation (Shridhar et al., 2023; James et al., 2022; Goyal et al., 2023; Gervet et al., 2023). Policies that use such 2D-to-BEV or 2D-to-3D scene encodings generalize better than their 2D counterparts and can handle novel camera viewpoints at test time. Interestingly, this improved performance is related to the way these methods handle multimodality in action prediction. 2D policy formulations, such as, RT-1 (Brohan et al., 2022), RT-2 (Brohan et al., 2023), Hiveformer (Guhur et al., 2022), InstructRL (Liu et al., 2022a) discretize the action dimensions directly, such as 3D translation and 3D rotation angles, and score bins using pooled 2D image features. BEV or 3D policy formulations such as Transporter Networks (Zeng et al., 2021), C2F-ARM (James et al., 2022), PerAct (Shridhar et al., 2023), Act3D (Gervet et al., 2023) and Robot view transformer (Goyal et al., 2023) instead discretize the robot's 3D workspace for localizing the robot's end-effector, such as the 2D BEV map for pick and place actions (Zeng et al., 2021), the 3D workspace (Gervet et al., 2023; Shridhar et al., 2023), or multiview re-projected images (Goyal et al., 2023). This results in spatially disentangled features that generalize better thanks to locality of computation: when the input scene changes, the output predictions change only locally (Zeng et al., 2021). Such 3D policies have not been combined yet with diffusion objectives.

In this paper, we propose 3D Diffuser Actor, a model that marries these two lines of work, diffusion policies for handling action multimodality and 3D scene encodings for effective spatial reasoning. 3D Diffuser Actor trains a denoising neural network that takes as input a 3D scene visual encoding, the current estimate of the end-effector's 3D location and orientation, as well as the iteration index, and predicts the error in 3D translation and rotation. Our model achieves translation invariance in prediction by "grounding" the current estimate of the robot's end-effector in the 3D scene workspace and featurizing them jointly using relative-position 3D attentions (Shaw et al., 2018; Su et al., 2021), as shown in Figure 1. Our output to input grounding operation relates our model to early work on iterative error feedback for structured prediction (Carreira et al., 2016), which similarly used an multi-step inference process and would render output human pose estimates onto the input image and jointly featurize them using rhetinotopic representations.

We test 3D Diffuser Actor in learning robot manipulation policies from demonstrations on the simulation benchmarks of RLbench (James et al., 2020) and in the real world. Our model sets a new state of the art on RLbench and outperforms existing 3D policies and 2D diffusion policies. It outperforms the previous SOTA with a 12% absolute gain. We ablate design choices of our model such as 3d scene encodings and 3D relative transformers for translation invariance, and find them all important in performance. Our models and code will be publicly available upon publication.

## 2 RELATED WORK

**Learning robot manipulation from demonstrations**   Though state to action mappings are typically multimodal, earlier works on learning from demonstrations train deterministic policies with behaviour cloning (Pomerleau, 1989; Bojarski et al., 2016). To better handle action multimodality, other approaches discretize action dimensions and use cross entropy losses (Lynch & Sermanet, 2020; Shridhar et al., 2023; Zeng et al., 2021). However, the number of bins needed to approximate a continuous action space grows exponentially with increasing dimensionality. Generative adversarial networks (Ho & Ermon, 2016; Tsurumine & Matsubara, 2022; Ding et al., 2019), variational autoencoders (Mandlekar et al., 2019) and combined Categorical and Gaussian distributions (Shafiullah et al., 2022; Guhur et al., 2022) have been used to learn from multimodal demonstrations. Nevertheless, these models tend to be sensitive to hyperparameters, such as the number of clusters used (Shafiullah et al., 2022). Implicit behaviour cloning represents distributions over actions by using Energy-Based Models (EBMs) (Florence et al., 2021; Ta et al., 2022). Optimization through EBMs amounts to searching the energy landscape for minimal-energy action, given a state. EBMs are inherently multimodal, since the learned landscape can have more than one minima. EBMs are also composable, which makes them suitable for combining action distributions with additional constraints during inference. Diffusion models (Sohl-Dickstein et al., 2015; Ho et al., 2020) are a powerful class of generative models related to EBMs in that they model the score of the distribution, else, the gradient of the energy, as opposed to the energy itself (Singh et al., 2023; Salimans & Ho, 2021). The key idea behind diffusion models is to iteratively transform a simple prior distribution into a target distribution by applying a sequential denoising process. They have been used for modeling state conditioned action distributions in imitation learning (Ryu et al., 2023; Urain et al., 2023; Pearce et al., 2023; Wang et al., 2022; Reuss et al., 2023; Mishra & Chen, 2023) from low dimensional input, as well as from visual sensory input, and show both better mode coverage and higher fidelity in action prediction than alternatives. They have not been yet combined with 3D scene representations.

**Diffusion models in robotics**   Beyond policy representations in imitation learning, diffusion models have been used to model cross-object and object-part arrangements (Liu et al., 2022b; Simeonov et al., 2023; Liu et al., 2023; Mishra & Chen, 2023; Fang et al., 2023; Gkanatsios et al., 2023a), and visual image subgoals (Kapelyukh et al., 2023; Dai et al., 2023; Ajay et al., 2023). They have also been used successfully in offline reinforcement learning (Chen et al., 2023a; Chi et al., 2023; Hansen-Estruch et al., 2023), where they model the state-conditioned action trajectory distribution (Hansen-Estruch et al., 2023; Chen et al., 2023a) or state-action trajectory distribution (Janner et al., 2022; He et al., 2023). ChainedDiffuser (Xian et al., 2023) proposes to forego motion planners commonly used for 3D keypose to 3D keypose linking, and instead used a trajectory diffusion model that conditions on the 3D scene feature cloud and the target 3D keypose to predict a trajectory from the current to the target keypose. 3D Diffuser Actor instead predicts the next 3D keypose for the robot's end-effector using 3D scene-conditioned diffusion, which is a much harder task than linking two given keyposes. ChainedDiffuser is complementary to our work: We can use keyposes predicted by 3D Diffuser Actor and link them with ChainedDiffuser's trajectories instead of motion planners. For the tasks in the Per-Act benchmark that we consider motion planner already work well, and thus we did not consider it for comparison.   Lastly, image diffusion models have been used for augmenting the conditioning images input to robot policies to help the latter generalize better (Liang et al., 2023; Chen et al., 2023b; Mandi et al., 2022).

**2D and 3D scene representations for robot manipulation**   End-to-end image-to-action policy models, such as RT-1 (Brohan et al., 2022), RT-2(Brohan et al., 2023), GATO (Reed et al., 2022), BC-Z (Jang et al., 2022), and InstructRL (Liu et al., 2022a), leverage transformer architectures for the direct prediction of 6-DoF end-effector poses from 2D video input. However, this approach comes at the cost of requiring thousands of demonstrations to implicitly model 3D geometry and adapt to variations in the training domains. Another line of research is centered around Transporter networks(Zeng et al., 2021; Seita et al., 2021; Shridhar et al., 2022; Gkanatsios et al., 2023a), demonstrating remarkable few-shot generalization by framing end-effector pose prediction as pixel classification. Nevertheless, these models are confined to top-down 2D planar environments with simple pick-and-place primitives. Direct extensions to 3D, exemplified by C2F-ARM (James et al., 2022) and PerAct (Shridhar et al., 2023), involve voxelizing the robot's workspace and learning to identify the 3D voxel containing the next end-effector keypose. However, this becomes computa-

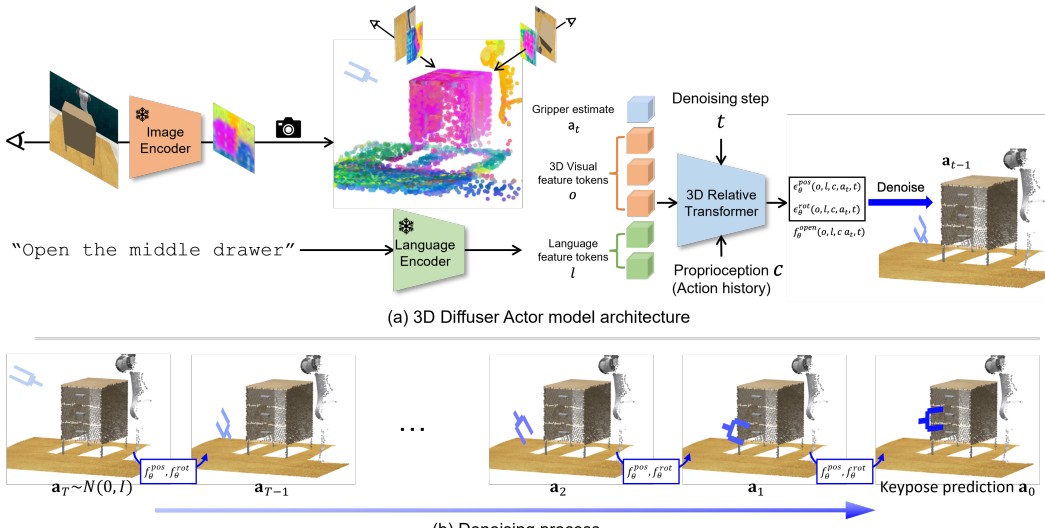

(a) 3D Diffuser Actor model architecture

(b) Denoising process

Figure 2: **3D Diffuser Actor architecture**. *Top:* 3D Diffuser Actor is a denoising diffusion probabilistic model of the robot 3D keyposes conditioned on sensory input, language instruction, action history and proprioceptive information. The model is a 3D relative position transformer that featurizes jointly the scene and the current noisy estimate for the pose of the robot's end-effector through 3D relative-position attentions. 3D Diffuser Actor outputs position and rotation residuals for denoising, as well as the end-effector's state (open/close). *Bottom:* 3D Diffuser Actor iteratively denoises the gripper pose estimate to target pose, where the pose estimate is initialized from pure noise.

tionally expensive as resolution requirements increase. Consequently, related approaches resort to either coarse-to-fine voxelization or efficient attention operations (Jaegle et al., 2021) to mitigate computational costs. Recently, Act3D (Gervet et al., 2023) uses coarse-to-fine 3D attentions in a 3D scene feature cloud and avoids 3D voxelization altogether. It samples 3D points in the empty workspace and featurizes them using cross-attentions to the physical 3D point features. Robot view transformer (RVT) (Goyal et al., 2023) re-projects the input RGB-D image to alternative image views, featurizes those and lifts the predictions to 3D to infer 3D locations for the robot's end-effector. Both Act3D and RVT show currently the highest performance on RLbench (James et al., 2020), the largest benchmark for learning from demonstrations. ~~3D Diffuser Actor achieves higher generalization due to grounding the end-effector poses as scene entities, thus enabling efficient relative attention.~~

## 3 3D DIFFUSER ACTOR FORMULATION

The architecture of 3D Diffuser Actor is shown in Figure 2. 3D Diffuser Actor formulates 3D robot action prediction as a conditional action distribution learning problem. It is a conditional diffusion model that takes as input visual observations, a language instruction, a short history trajectory of the end-effector and the current estimate for the robot's future action, and predicts the error in 3D translation and 3D orientation. We review Denoising Diffusion Probabilistic models in Section 3.1 and describe the architecture and training details of our model in Section 3.2.

### 3.1 DENOISING DIFFUSION PROBABILISTIC MODELS

A diffusion model learns to model a probability distribution $p(x)$ by inverting a process that gradually adds noise to a sample $x$. For us, $x$ represents 3D translations and 3D rotations of the end-effector's next keypose. The diffusion process is associated with a variance schedule $\{\beta_t \in (0, 1)\}_{t=1}^{T}$, which defines how much noise is added at each time step. The noisy version of sample $x$ at time $t$ can then be written $x_t = \sqrt{\bar{\alpha}_t}x + \sqrt{1 - \bar{\alpha}_t}\epsilon$ where $\epsilon \sim \mathcal{N}(\mathbf{0}, \mathbf{1})$, is a sample from a Gaussian distribution (with the same dimensionality as $x$), $\alpha_t = 1 - \beta_t$, and $\bar{\alpha}_t = \prod_{i=1}^{t} \alpha_i$. The denoising process is modeled by a neural network $\hat{\epsilon} = \epsilon_\theta(x_t; t)$ that takes as input the noisy sample $x_t$ and the noise level $t$ and tries to predict the noise component $\epsilon$.

Diffusion models can be easily extended to draw samples from a distribution $p(x|\mathbf{c})$ conditioned on input $\mathbf{c}$, which is added as input to the network $\epsilon_\theta$. For us $\mathbf{c}$ is the visual scene captured by one or more calibrated RGB-D images, a language instruction, as well as a short history of the robot's end-effector.

Given a collection of $\mathcal{D} = \{(x^i, \mathbf{c}^i)\}_{i=1}^N$ of end-effector keyposes $x^i$ paired with observation and robot history context $\mathbf{c}^i$, the denoising objective becomes:

$$\mathcal{L}_{\text{diff}}(\theta; \mathcal{D}) = \frac{1}{|\mathcal{D}|} \sum_{x^i, \mathbf{c}^i \in \mathcal{D}} ||\epsilon_\theta(\sqrt{\bar{\alpha}_t} x^i + \sqrt{1 - \bar{\alpha}_t}\epsilon, \mathbf{c}^i, t) - \epsilon||. \tag{1}$$

This loss corresponds to a reweighted form of the variational lower bound for $\log p(x|\mathbf{c})$ (Ho et al., 2020).

In order to draw a sample from the learned distribution $p_\theta(x|\mathbf{c})$, we start by drawing a sample $x_T \sim \mathcal{N}(\mathbf{0}, \mathbf{1})$. Then, we progressively denoise the sample by iterated application of $\epsilon_\theta$ $T$ times according to a specified sampling schedule (Ho et al., 2020; Song et al., 2020), which terminates with $x_0$ sampled from $p_\theta(x)$:

$$x_{t-1} = \frac{1}{\sqrt{\alpha_t}} \left( x_t - \frac{\beta_t}{\sqrt{1 - \bar{\alpha}_t}} \epsilon_\theta(x_t, t, \mathbf{c}) \right) + \frac{1 - \bar{\alpha}_{t+1}}{1 - \bar{\alpha}_t} \beta_t \mathbf{z}, \text{ where } \mathbf{z} \sim \mathcal{N}(\mathbf{0}, \mathbf{1}) \tag{2}$$

### 3.2 3D DIFFUSER ACTOR

We next describe the architecture of our network that predicts the end-effector's pose error given the current estimate, a visual scene encoding, a language instruction, the denoising iteration index and the robot's history.

Instead of predicting an end-effector pose at each timestep, we extract a set of *keyposes* that capture bottleneck end-effector poses in a demonstration, following prior work (James & Davison, 2022; Shridhar et al., 2023; Guhur et al., 2023; Liu et al., 2022a). Following Act3D (Gervet et al., 2023), "a pose is a keypose if (1) the end-effector changes state (something is grasped or released) or (2) velocities approach near zero (a common occurrence when entering pre-grasp poses or entering a new phase of a task). The action prediction problem then boils down to predicting a plausible action keypose given the current robot state." At inference time, 3D Diffuser Actor iteratively denoises its predictions to infer the next 3D keypose. The predicted 3D keypose is reached with a sampling-based motion planner, following previous works (Shridhar et al., 2023; Guhur et al., 2023).

Following Act3D (Gervet et al., 2023), we assume access to a dataset of $N$ demonstration trajectories. Each demonstration is a sequence of observations $o_\tau = \{o_1, o_2, .., o_t\}$ paired with continuous actions $\mathbf{a}_\tau = \{\mathbf{a}_1, \mathbf{a}_2, .., \mathbf{a}_t\}$ and a language instruction $l$ that describes the task. Each observation $o_t$ consists of a set of posed RGB-D images taken from one or more camera views. An action $\mathbf{a}_t$ consists of the 3D position and 3D orientation of the robot's end-effector, as well as its binary open or closed state: $\mathbf{a} = \{\mathbf{a}^{\text{pos}} \in \mathbb{R}^3, \mathbf{a}^{\text{rot}} \in \mathbb{R}^6, \mathbf{a}^{\text{open}} \in \{0, 1\}\}$. We represent rotations using the 6D rotation representation of Zhou et al. (2020) for all environments in all our experiments, to avoid the discontinuities of the quaternion representation that previous approaches use (Guhur et al., 2023; Gervet et al., 2023).

**Scene and language encoder** We use a scene and language encoder similar to Act3D (Gervet et al., 2023). We describe it here to make the paper self-contained. Our scene encoder maps posed multi-view RGB-D images into a multi-scale scene feature cloud. We use a pre-trained 2D feature extractor followed by a feature pyramid network to extract multi-scale visual tokens from each camera view. We associate every 2D feature grid location in the 2D feature maps with a depth value, by averaging the depth values of the image pixels that correspond to it. Then we use the camera intrinsics and pinhole camera equation to map a pixel location and depth value $(x, y, d)$ to a 3D location $(X, Y, Z)$, and "lift" the 2D feature tokens to 3D to obtain a 3D feature cloud. The language encoder featurizes language task descriptions or instructions into language feature tokens. We use the pretrained CLIP ResNet50 2D image encoder (Radford et al., 2021) to encode each RGB image into a 2D feature map and pretrained CLIP language encoder to encode the language task instruction. We keep those frozen.

**3D grounding of the current action estimate** We disentangle "what" and "where" when representing the end-effector's action estimate. For "what", we encode the 3D pose estimate into a high-dimensional representation through an MLP. For "where", we treat the noisy estimate as a 3D entity rendered at the corresponding 3D location in the scene, as shown in Figure 2. This enables relative cross-attention with the scene and other physical entities, as we explain in the following.

We incorporate proprioceptive information as a short history of end-effector positions. These are represented as scene entities of known 3D positions while their features are learnable.

**3D Relative Position Diffusion Transformer**   We contextualize all tokens, namely visual $o$, language $l$, proprioception $c$ and current estimate tokens $\mathbf{a}_t^{pos}, \mathbf{a}_t^{rot}$ using 3D relative position attention layers. Inspired by recent work in visual correspondence (Li & Harada, 2022; Gkanatsios et al., 2023b) and 3D manipulation (Gervet et al., 2023), we use rotary positional embeddings (Su et al., 2021). These have the property that the dot product of two positionally encoded features $\mathbf{x}_i, \mathbf{x}_j$ is:

$$\mathbf{PE}(\mathbf{p}_i, \mathbf{x}_i)^T \mathbf{PE}(\mathbf{p}_j, \mathbf{x}_j) = \mathbf{x}_i^T \mathbf{M}(\mathbf{p}_i)^T \mathbf{M}(\mathbf{p}_j) \mathbf{x}_j = \mathbf{x}_i^T \mathbf{M}(\mathbf{p}_j - \mathbf{p}_i) \mathbf{x}_j \tag{3}$$

which depends only on the relative positions of points $\mathbf{p}_i$ and $\mathbf{p}_j$ and thus is translation-invariant. The updated feature token that corresponds to the current action estimate token is fed to MLPs to predict the position error $\epsilon_\theta^{\mathrm{pos}}(o, l, c, \mathbf{a}_t^{\mathrm{pos}}, \mathbf{a}_t^{\mathrm{rot}}, t)$, rotation error $\epsilon_\theta^{\mathrm{rot}}(o, l, c, \mathbf{a}_t^{\mathrm{pos}}, \mathbf{a}_t^{\mathrm{rot}}, t)$ and whether the gripper should be open or closed $f_\theta^{\mathrm{open}}(o, l, c, \mathbf{a}_t^{\mathrm{pos}}, \mathbf{a}_t^{\mathrm{rot}}, t)$.

**Denoising process**   We use a modified version of Equation 2 to update the current estimate of the end-effector's pose:

$$\mathbf{a}_{t-1}^{\mathrm{pos}} = \frac{1}{\sqrt{\alpha_t}} \left( \mathbf{a}_t^{\mathrm{pos}} - \frac{\beta_t}{\sqrt{1 - \bar{\alpha}_t}} \epsilon_\theta^{\mathrm{pos}}(o, l, c, \mathbf{a}_t^{\mathrm{pos}}, \mathbf{a}_t^{\mathrm{rot}}, t) \right) + \frac{1 - \bar{\alpha}_{t+1}}{1 - \bar{\alpha}_t} \beta_t \mathbf{z}^{pos} \tag{4}$$

$$\mathbf{a}_{t-1}^{\mathrm{rot}} = \frac{1}{\sqrt{\alpha_t}} \left( \mathbf{a}_t^{\mathrm{rot}} - \frac{\beta_t}{\sqrt{1 - \bar{\alpha}_t}} \epsilon_\theta^{\mathrm{rot}}(o, l, c, \mathbf{a}_t^{\mathrm{rot}}, \mathbf{a}_t^{\mathrm{rot}}, t) \right) + \frac{1 - \bar{\alpha}_{t+1}}{1 - \bar{\alpha}_t} \beta_t \mathbf{z}^{rot} \tag{5}$$

where $\mathbf{z}^{pos}, \mathbf{z}^{rot} \sim \mathcal{N}(\mathbf{0}, \mathbf{1})$ variables of appropriate dimension. We use the following two noise schedulers:

1. a scaled-linear noise scheduler $\beta_t = (\beta_{\max} - \beta_{\min})t + \beta_{\min}$, where $\beta_{\max}, \beta_{\min}$ are hyperparameters, set to 0.02 and 0.0001 in our experiments,

2. a squared cosine noise scheduler $\beta_t = \frac{1 - \cos\left(\frac{(t+1)/T + 0.008}{1.008} * \frac{\pi}{2}\right)^2}{\cos\left(\frac{t/T + 0.008}{1.008} * \frac{\pi}{2}\right)^2}$.

We found using a scale-linear noise schedule for denoising end-effector's 3D positions and a squared cosine noise schedule for denoising the end-effector's 3D orientations to converge much faster than using squared cosine noise for both. We justify the use of square cosine noise scheduler for 3D orientations in the Appendix.

**Training**   3D Diffuser Actor is trained on a dataset of kinesthetic demonstrations, which consists of tuples of RGB-D observations, proprioception information of robot's end-effector pose $c$, action trajectories $\mathbf{a} = [\mathbf{a}^{pos}, \mathbf{a}^{rot}]$, and corresponding language goals $\mathcal{D} = \{(o_1, c_1, \mathbf{a}_1, l_1), (o_2, c_2, \mathbf{a}_2, l_2), ...\}$. During training, we randomly sample a diffusion step $t$ and add noise $\epsilon = (\epsilon_t^{\mathrm{pos}}, \epsilon_t^{\mathrm{rot}})$ to the ground-truth target action. We supervise 3D Diffuser Actor using a denoising objective: conditioning the tuple of $(o, l, c, t)$, our model learns to reconstruct the clean action by predicting the pose transformation wrt the current estimate. We adopt the $L1$ loss for reconstructing the 3D position and 3D rotation error. We use binary cross-entropy loss to supervise end-effector opening. Our objective reformulates Equation 1 into:

$$\mathcal{L}_\theta = \frac{1}{|\mathcal{D}|} \sum_{i=1}^{|\mathcal{D}|} \mathrm{BCE}(f_\theta^{\mathrm{open}}(o_i, l_i, c_i, \mathbf{a}_{t,i}^{\mathrm{pos}}, \mathbf{a}_{t,i}^{\mathrm{rot}}, t), \mathbf{a}_i^{\mathrm{open}}) +$$

$$w_1 \cdot \|(\epsilon_\theta^{\mathrm{pos}}(o_i, l_i, c_i, \mathbf{a}_{t,i}^{\mathrm{pos}}, \mathbf{a}_{t,i}^{\mathrm{rot}}, t) - \epsilon_t^{\mathrm{pos}}\| + w_2 \cdot \|(\epsilon_\theta^{\mathrm{rot}}(o_i, l_i, c_i, \mathbf{a}_{t,i}^{\mathrm{pos}}, \mathbf{a}_{t,i}^{\mathrm{rot}}, t) - \epsilon_t^{\mathrm{rot}}\|, \tag{6}$$

where $w_1, w_2$ are hyperparameters estimated using cross-validation.

**Implementation details**   We render images at $256 \times 256$ resolution. Based on the depth and camera parameters, we calculate xyz-coordinates of each image pixel and use such information to lift 2D image features to 3D. To reduce the memory footprint in our 3D Relative Transformer, we use Farthest Point Sampling to sample $20\%$ of the points in the input 3D feature cloud. We use FiLM (Perez et al., 2018) to inject conditional input, including the diffusion step and proprioception history, to

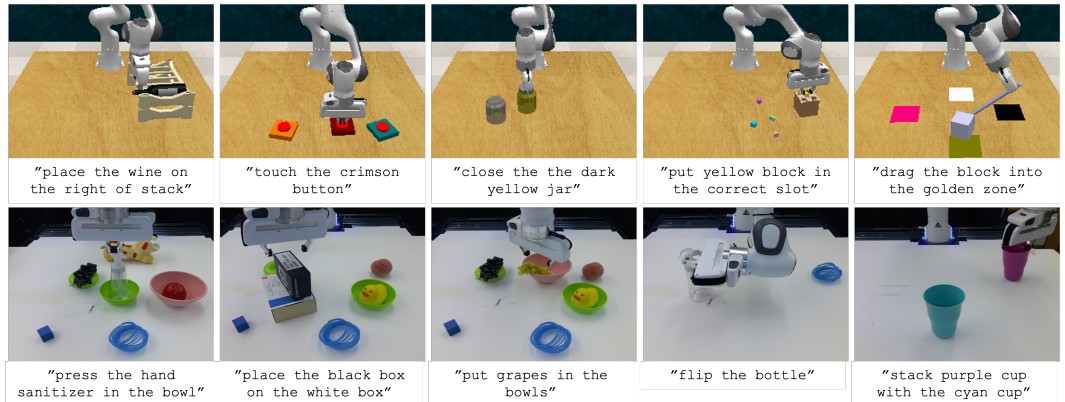

Figure 3: **Tasks.** We conduct experiments on 52 simulated tasks in RLBench (James et al., 2020) (only 5 shown), and 5 real-world tasks. Please see the supplementary videos for results of our model in the real world.

every attention layer in the model. We include a detailed architecture diagram of our model in the Appendix. We augment RGB-D observations with random rescaling and cropping. Nearest neighbor interpolation is used for rescaling RGB-D observations. During training, we use a batch size of 48, embedding dimension of 120, and training iterations of 600K for the 18 PerAct-task experiments. We set the batch size to 32, embedding dimension to 60, and training steps to 200K for the ablations and real-world experiments. We set the loss weights to $w_1 = 100$ and $w_2 = 10$. During testing, we use a low-level motion planner to reach predicted keyposes. We use the native motion planner–BiRRT (Kuffner & LaValle, 2000) in RLBench. For real-world experiments, we use the same BiRRT planner provided by the MoveIt! ROS package (Coleman et al., 2014).

## 4 EXPERIMENTS

We test 3D Diffuser Actor' ability to learn manipulation policies from demonstrations in simulation and in the real-world. Our experiments aim to answer the following questions:

**1.** How does 3D Diffuser Actor compare against the state-of-the-art manipulation policies in generalizing across diverse tasks and environments?

**2.** How does 3D Diffuser Actor compare against 2D policy diffusion models, that use 2D multiview encodings of the scene instead of 3D ones?

**3.** How does architectural choices, such as 3D relative attentions and noise schedule selection impact performance?

**Baselines** We compare 3D Diffuser Actor to the following state-of-the-art manipulation policy learning methods: **1.** InstructRL (Liu et al., 2022a), a 2D policy that directly predicts 6 DoF poses from image and language conditioning with a pre-trained vision-and-language backbone. **2.** PerAct (Shridhar et al., 2023), a 3D policy that voxelizes the workspace and detects the next best voxel action through global self-attention. **3.** Act3D (Gervet et al., 2023), a 3D policy that featurizes the robot's 3D workspace using coarse-to-fine sampling and featurization. **4.** RVT (Goyal et al., 2023), a 3D policy that deploys a multi-view transformer to predict actions and fuse those across views by back-projecting back to 3D. **5.** *2D Diffuser Actor*, our implementation of (Wang et al., 2022). We remove the 3D scene encoding from 3D Diffuser Actor, and instead generate per-image 2D representations by average-pooling features within each view, following (Chi et al., 2023). We add learnable embeddings to distinguish different views. We use standard attention layers for joint encoding the action estimate and 2D image features. **6.** *3D Diffuser Actor -RelAtt*, an ablative version of our model that uses standard non-relative attentions to featurize the current rotation and translation estimate with the 3D scene feature cloud, and does not ground the current gripper estimate in the scene. This version of our model is not translation-invariant. For InstructRL, PerAct, Act3D and RVT we report the results from the corresponding papers.

| method | Avg. Success ↑ | Avg. Rank ↓ | open drawer | slide block | sweep to dustpan | meat off grill | turn tap | put in drawer | close jar | drag stick |
|---|---|---|---|---|---|---|---|---|---|---|
| PerAct (Shridhar et al., 2023) | 43 | 3.1 | 80 | 72 | 56 | 84 | 80 | 68 | 60 | 68 |
| RVT (Goyal et al., 2023) | 63 | 2.7 | 71 | 82 | 72 | 88 | 94 | 88 | 52 | 99 |
| Act3D (Gervet et al., 2023) | 65 | 2.4 | 93 | 93 | 92 | 94 | 94 | 90 | 92 | 92 |
| 3D Diffuser Actor (ours) | **77 (+12)** | 1.4 | 87 (-6) | **97 (+4)** | **99 (+7)** | **95 (+1)** | **97 (+3)** | **92 (+2)** | 53 (-39) | **100 (+1)** |

| method | stack blocks | screw bulb | put in safe | place wine | put in cupboard | sort shape | push buttons | insert peg | stack cups | place cups |
|---|---|---|---|---|---|---|---|---|---|---|
| PerAct (Shridhar et al., 2023) | 36 | 24 | 44 | 12 | 16 | 20 | 48 | 0 | 0 | 0 |
| RVT (Goyal et al., 2023) | 29 | 48 | 91 | 91 | 50 | 36 | 100 | 11 | 26 | 4 |
| Act3D (Gervet et al., 2023) | 12 | 47 | 95 | 80 | 51 | 8 | 99 | 27 | 9 | 3 |
| 3D Diffuser Actor (ours) | **68 (+32)** | **91 (+43)** | **99 (+4)** | 90 (-1) | **74 (+23)** | 1 (-35) | 99 (-1) | **79 (+52)** | **34 (+8)** | **32 (+28)** |

Table 1: **Multi-Task Performance on RLBench.** We report success rates on 18 RLBench tasks with 249 variations. Our 3D Diffuser Actor outperforms prior state-of-the-art baselines–PerAct, RVT, and Act3D–among most tasks by a large margin.

**Evaluation metrics**   Following previous work (Gervet et al., 2023; Shridhar et al., 2023),   we evaluate policies by task completion success rate, the proportion of execution trajectories that lead to goal conditions specified in language instructions.

| Method | Avg. Success |
|---|---|
| 3D Diffuser Actor | 68.5 |
| 2D Diffuser Actor | 40 |
| 3D Diffuser Actor -RelAtt | 62 |

Table 2: **Ablations.** We evaluate all policies on 5 Hiveformer tasks under multi-task settings. Both 3D scene encodings and 3D relative attentions matter in performance.

| 3D Diffuser Actor | Act3D |
|---|---|
| 3 secs | 0.12 secs |

Table 3: **Inference time per keypose.**

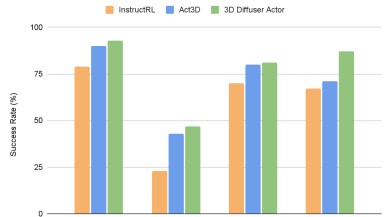

Figure 4: **Single-task performance.**   On 34 tasks, spanning four different categories, 3D Diffuser Actor outperforms prior state-of-the-art models InstructRL and Act3D.

## 4.1 EVALUATION IN SIMULATION

**Datasets**   We test 3D Diffuser Actor on RLBench in a multi-task multi-variation setting.  The benchmark is build atop CoppeliaSim (Rohmer et al., 2013) simulator, where a Franka Panda Robot is used to complete the tasks. We use four cameras (*front*, *wrist*, *left shoulder*, and *right shoulder*) to capture RGB-D visual observations, in accordance to previous works (Gervet et al., 2023). We test our model and baselines on a suite of 18 tasks over total 249 variations, initially introduced in (Shridhar et al., 2023). Each task has 2-60 variations, which differ in object pose, appearance, and semantics. These tasks are specified by language descriptions. We train our model and baselines using 100 demonstrations per task, which are evenly split across variations, and test them on 100 unseen episodes for each task. During evaluation, we allow models to predict and execute a maximum of 50 actions unless they receive earlier task-completion indicators from the simulation environment.

We show quantitative results for multi-task manipulation in Table 1. Our 3D Diffuser Actor outperforms Act3D, RVT, and PerAct on most tasks with a large margin. It achieves an average 77% success rate, an absolute improvement of 12% over Act3D, the previous state-of-the-art. In particular, 3D Diffuser Actor excels at tackling tasks with multiple modes, such as *stack blocks*, *stack cups*, and *place cups*, which most baselines fail to complete. We obtain substantial improvements of +32%, +43%, +23%, +52%, and +28% on *stack blocks*, *screw bulb*, *put in cupboard*, *insert peg*, and *place cups* tasks.

We additionally show single-task results in Figure 4 on 34 selected tasks from the setup of Guhur et al. (2023), spanning four categories (*multimodal*, *long-term*, *visual occlusion*, and *tools*). 3D Diffuser Actor outperforms both InstructRL and Act3D on all tested categories by an average absolute margin of 6%.

| Task | Success |
|---|---|
| pick bowl | 100 |
| stack bowl | 100 |
| put grapes | 50 |
| fold towel | 70 |
| press sanitizer | 100 |

Table 4: **Single-Task Performance on real-world tasks.**

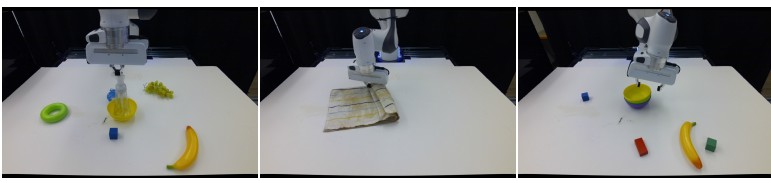

"press hand the sanitizer"  "fold the towel"  "stack to the purple bowl"

Figure 5: **Visualized results of our real-world manipulation.**

## 4.2 ABLATIONS

We ablate different design choices of 3D Diffuser Actor, namely, the use of 3D scene representations and relative attentions. We train multi-task policies on 5 tasks of RLBench proposed in HiveFormer (Guhur et al., 2023): *reach_and_drag*, *hang_frame*, *slide_cabinet*, *stack_cups*, and *open_fridge*.

We show the ablation results in Table 2. We draw the following conclusions:

**2D vs. 3D Diffuser Actor:** 3D Diffuser Actor largely outperforms its 2D counterpart where the input images are featurized, pooled and fed into the model.

**3D relative attentions:** Our model with absolute attentions performs worse than our model with relative attentions, showing that translation invariance is important for generalization.

## 4.3 EVALUATION IN THE REAL WORLD

We validate 3D Diffuser Actor in learning manipulation tasks from real world demonstations. We use a Franka Emika robot and capture visual observations with a Azure kinect RGB-D sensor at a front view. Images are originally captured at $1280 \times 720$ resolution, and downsampled to a resolution of $256 \times 256$. Camera extrinsics are calibrated w.r.t the robot base. We choose five tasks–*pick a bowl*, *stack bowls*, *put grapes in bowls*, *fold a towel*, and *press sanitizer*. We collect 20 keypose demonstrations per task. The evaluation metric is average success rate of each task.

Our 3D Diffuser Actor successfully solves the real-world tasks (Table 4). In particular, we show in Fig 5 that our model is able to predict different modes of manipulation policies for the similar scene configuration. For example, the model can pick up a bowl with 4 different poses or put different grapes into the bowl. Video results for these and other tasks are available at our project page https://sites.google.com/view/3d-diffuser-actor.

## 4.4 LIMITATIONS

Our framework currently has the following limitations: **1.** 3D Diffuser Actor requires multiple iterations of denoising, which results in higher inference latency than non-diffusion baselines. **2.** Our model conditions on 3D scene representations, which require camera calibration and depth information. **3.** 3D Diffuser Actor requires supervision from kinesthetic demos which are overall hard to collect. Combining such supervision with reinforcement learning in simulation is an exiting future direction. **4.** 3D Diffuser Actor considers only visual sensory input. Using audio or tactile input or force feedback would make our model's predictions more robust. **5.** All tasks in RLbench are quasi-static. Extending our method to dynamic tasks is a direct avenue of future work.

## 5 CONCLUSION

We present 3D Diffuser Actor, a 3D robot manipulation policy that uses generative diffusion objectives. Our method sets a new-state-of-the-art on RLbench outperforming both existing 3D policies and 2D diffusion policies. We introduce important architectural innovations, such as 3D grounding of the robot's estimate and 3D relative position transformers that render 3D Diffuser Actor translation invariant, and empirically verify their contribution to performance. We further test out model in the real world and show it can learn to perform multimodal manipulation tasks from a handful of demonstrations. Our future work will attempt to train such 3D Diffuser Actor policies large scale in domain-randomized simulation environments, to help them transfer from simulation to the real world.

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

## A APPENDIX

We present a detailed version of our architecture in Section A.1. We visualize the importance of a square cos variance scheduler for denoising the rotation estimate in Section A.2.

### A.1 DETAILED MODEL DIAGRAM

We present a more detailed architecture diagram of our 3D Diffuser Actor in Figure 6.

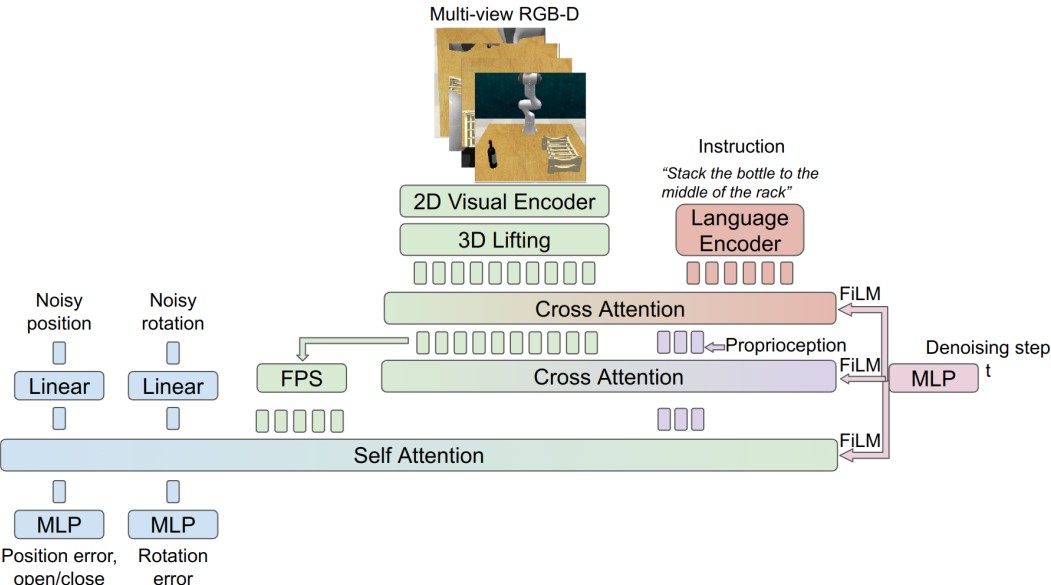

Figure 6: **3D Diffuser Actor architecture** in more detail.

The inputs to our network are i) a stream of RGB-D views; ii) a language instruction; iii) proprioception in the form of end-effector's history poses; iv) the current noisy estimates of position and rotation; v) the denoising step $t$. The images are encoded into visual tokens using a pretrained 2D backbone. The depth values are used to "lift" the multi-view tokens into a 3D feature cloud. The language is encoded into feature tokens using a language backbone. The proprioception is represented as learnable tokens with known 3D locations in the scene. The noisy estimates are fed to linear layers that map them to high-dimensional vectors. The denoising step is fed to an MLP.

The visual tokens cross-attend to the language tokens and get residually updated. The proprioception tokens attend to the visual tokens to contextualize with the scene information. We subsample a number of visual tokens using Farthest Point Sampling (FPS) in order to decrease the computational requirements. The sampled visual tokens, proprioception tokens and noisy position/rotation tokens attend to each other. We modulate the attention using adaptive layer normalization and FiLM (Perez et al., 2018). Lastly, the contextualized noisy estimates are fed to MLP to predict the error terms as well as the end-effector's state (open/close).

### A.2 THE IMPORTANCE OF NOISE SCHEDULER

We visualize the clean/noised 6D rotation representations as two three-dimensional unit-length vectors in Figure 7. We plot each vector as a point in the 3D space. We can observe that noised rotation vectors generated by the squared linear scheduler cover the space more completely than those by the scaled linear scheduler.

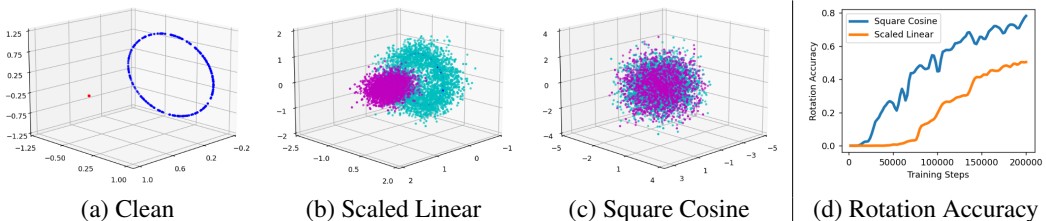

(a) Clean      (b) Scaled Linear      (c) Square Cosine      (d) Rotation Accuracy

Figure 7: **V**isualization of noised rotation based on different schedulers. We split the 6 DoF rotation representations into 2 three-dimension unit-length vectors, and plot the first/second vector as a point in 3D. The noised counterparts are colorized in magenta/cyan. We visualize the rotation of all keyposes in RLBench *insert_peg* task. From left to right, we visualize the (a) clean rotation, (b) noisy rotation with a scaled-linear scheduler, and (c) that with a square cosine scheduler. Lastly, we compare (d) the denoising performance curve of two noise schedulers. Using the square cosine scheduler helps our model to denoise from the pure noise better.

