# OpenReview forum: "3D Diffuser Actor: Multi-task 3D Robot Manipulation with Iterative Error Feedback"
_ICLR.cc/2024/Conference — Submitted to ICLR 2024_

### Official Review · Reviewer_sG3K · 2023-10-29

**Soundness:** 2 fair
**Presentation:** 3 good
**Contribution:** 3 good
**Rating:** 5
**Confidence:** 4

**Summary:**

This paper proposes 3D Diffuser Actor, a new behavior cloning algorithm combining diffusion policy and 3D representations for multi-task robotic manipulation. Utilizing the 3D representation and the attention mechanism, the proposed method achieves new SOTA on RLBench tasks. The authors also conduct real robot experiments with the newly proposed method, showing the applicability of the diffusion-based actor.

**Strengths:**

- **Motivation is good and natural**. Diffusion policies have achieved success in fitting distributions, and introducing them into 3D is a necessary step.
- **Good results and extensive experiments.** The results (12\% improvements) seem to be significant, which are gained on a multi-task benchmark across diverse tasks, and showing some real robot experiments are also very necessary for such robotic manipulation agents.

**Weaknesses:**

- **No deeper analysis about why diffusion models could help**. The ablation results only show two factors matter, but all these factors seem to be not novel and not surprising, thus deeper analysis might be necessary. I have also checked the supplementary files and the presented Figure 7 looks interesting, but why `scaled linear` is worse than `square cosine` when the former one seems to cover the original distribution better?
- **The inference time and the denoising steps are both not clear.** The proposed method achieves 12% absolute gain, but considering the inference time of diffusion models, this gain might be not obvious in the real world, due to huge latency. Could the authors also report the wall time between different algorithms?
- **The evaluation process is not clear**. Is the result in Table 1 the best success rate over a lot of checkpoints? And how many training
epochs are used? How many episodes are tested during the evaluation? How many seeds are used for the main results (Table 1)?
- **Lack of baselines in real robot experiments**.
- **Lack of discussion and experiment comparison with recent related works such as GNFactor [2].** I think this very recent method [1] could possibly serve as a baseline and it would be good to see some direct experiment results.
- **Lack of multi-task manipulation results in real robot experiments**. Both PerAct [1] and GNFactor [2] have shown ability to execute real-world multi-task manipulation, and it could be good to compare the multi-task performance in real robot also, considering this work is closely related to PerAct [1] and GNFactor [2].
- **Typo** in Figure 2 (a): Acter -> Actor

Overall, I tend to reject this paper with a score slightly lower than borderline,  considering the above issues for the initial review.  I would carefully consider raising my score if my questions are well addressed.

[1] Shridhar, Mohit, et al. "Perceiver-actor: A multi-task transformer for robotic manipulation." CoRL, 2022.

[2] Ze, Yanjie, et al. "Gnfactor: Multi-task real robot learning with generalizable neural feature fields." CoRL, 2023.

**Questions:**

See `weakness` above.

---

> ### Comment · Reviewer_sG3K · 2023-11-13
> **Update after reading the review from  Reviewer Ws6y**
>
> I also found that this paper is very similar to Act3D (it seems a lot of contents are borrowed). In addition, this work [1] seems to be also very related and in a very close style.
>
> I personally think this problem might be **much worse** than just a strong reject, if without any descent explanation from the authors.
>
> [1] ChainedDiffuser: Unifying Trajectory Diffusion and Keypose Prediction for Robotic Manipulation, https://chained-diffuser.github.io/

---

> ### Author Response · Authors · 2023-11-15
> **Authors' Response**
>
> ### 1. Further analysis on the importance of diffusion
> >No deeper analysis about why diffusion models could help. The ablation results only show two factors matter, but all these factors seem to be not novel and not surprising, thus deeper analysis might be necessary.
>
>
> Motivated by your comment and the comment of Reviewer Ws6y, and in an attempt to validate the argument regarding utility for diffusion, we conducted an additional experiment where we use a close to **identical** architecture to 3D Diffuser Actor for **regressing** the 3D location and 3D rotation of the target keypose, as opposed to the noise and the use of iterative denoising (no denoising iteration index input for  regression). We show the comparison for 3D location prediction on train and val sets ([here](https://docs.google.com/presentation/d/e/2PACX-1vTZkgzAxzw2IlVY86Ft9AmRC2UR_RBVFV8uFBdeyxeg9hdCWN7mstblmIkOQAogGR2LT1k3dsLEXcTo/pub?start=false&loop=false&delayms=3000)) and the comparison for 3D rotation prediction on train and val sets ([here](https://docs.google.com/presentation/d/e/2PACX-1vSkoF10a1x7D12Ll1xCeJ4rPSA02GzTKPsCq0yS8dJWsZYuUnFAGHjCixD_blR1_hyDBdluIqTtpbsF/pub?start=false&loop=false&delayms=3000#slide=id.g29b18ed0d7c_0_0)).
>
> As you can see, 3D diffuser actor generalizes much better than the regression model, especially for 3D location.  Both models use the same scene encoder, language encoder, and 3D relative transformer decoder architecture, but one predicts the end-effector through iterative denoising, while the other(regression) through a single feedforward pass.
> **Our contribution**: We propose the first 3D keypose diffusion model denoised from 3D scene representations for learning 3D manipulation from demonstrations. We achieve superior performance over many prior methods in apples-to-apples comparisons in an established experimental setup on RLBench(+12% compared to state-of-the-art methods).   Our model, 3D Diffuser Actor, marries 3D representations with diffusion policies for robot manipulation, as  we describe  clearly in the abstract (*"a framework that marries diffusion policies and 3D scene representations for robot manipulation"*) and in the intro. In both abstract and intro  we first describe diffusion policies (in abstract: *"Diffusion policies capture the action distribution conditioned on the robot and environment state using conditional diffusion models. They have recently shown to outperform both deterministic and alternative generative policy formulations in learning from demonstrations."*), then 3D policies (in abstract: *"3D robot policies use 3D scene feature representations aggregated from single or multiple 2D image views using sensed depth. They typically generalize better than their 2D counterparts in novel viewpoints."*) and we say we introduce 3D diffusion policies that marry the above two: (in abstract: *"We unify these two lines of work and present a neural policy architecture that uses 3D scene representations to iteratively denoise robot 3D rotations and translations for a given language task description."*)
> **The specific way this marriage is accomplished is important**:  we represent the current noisy end-effector pose estimate as a 3D token in the 3D feature cloud and featurize it  with relative 3D attentions. Since the ICLR submission deadline, **we ran the exact same model with absolute position attention (closer to what 2D diffusion policies do) on the whole PerAct benchmark and obtained an 8% worse performance**. We consider the above 3D diffusion keypose policy model to be our contribution.
>
> **Please, also see the general response in the beginning of the rebuttal.**
>
>
>
> >I have also checked the supplementary files and the presented Figure 7 looks interesting, but why scaled linear is worse than square cosine when the former one seems to cover the original distribution better?
>
> The objective of the noising process is to cover the whole space densely, as during inference we start from pure Gaussian noise and gradually denoise, following the DDPM method. As Figure 7 shows, the scaled linear schedule captures the distribution instead of covering the whole space of possible rotations. As a result, when randomly initializing from Gaussian noise, our model gets out of distribution and cannot denoise to a valid solution.

---

> > ### Author Response · Authors · 2023-11-15
> > **Authors' Response (2)**
> >
> > ### 2. Inference time
> > >The inference time and the denoising steps are both not clear. The proposed method achieves 12% absolute gain, but considering the inference time of diffusion models, this gain might be not obvious in the real world, due to huge latency. Could the authors also report the wall time between different algorithms?
> >
> > Right below we report wall time using a Nvidia GeForce GTX 1080 graphic card.  We use 100 diffusion steps during both training and testing. As we can see, our method is much much slower than Act3d. We are currently working on accelerating inference. The inference time can be drastically reduced using 1) fewer denoising steps, we  suspect that 15 steps may suffice for equivalent performamce based on recent literature (Reuss et al, RSS 2023)  2) more efficient attention implementation (e.g., flash attention [Dao et al, NeuRIPS 2022]). 3) faster diffusion process (e.g. Consistency Model [Song et al, ICML 2023]). We have added the inference time in our manuscript.
> >
> >
> > ||3D Diffuser Actor | Act3D |
> > |:---------------:|:---------------:|:---------------:|
> > |Inference Time per keypose  |3 secs | 0.12 secs|
> >
> > ### 3. Evaluation details
> > >The evaluation process is not clear. Is the result in Table 1 the best success rate over a lot of checkpoints? And how many training epochs are used? How many episodes are tested during the evaluation? How many seeds are used for the main results (Table 1)
> >
> > We evaluate the last checkpoint of training using one random seed.
> >
> > We follow PerAct / HiveFormer to use 100 episodes per task for training and testing.  See “Datasets” subsection.
> >
> > We have detailed the hyper-parameters of training in the “implementation details” subsection.  Specifically, we train the model with 600K  for the 18 PerAct-task  and  200K iterations for the ablation experiments.
> >
> > ### 4. Real-robot experiments
> > >Lack of baselines in real-robot experiments.
> >
> > This is hard to do as it requires resetting the real environment to  identical state across multiple models. PerAct, Act3D, RVT, InstructRL and most other works  do not consider baselines in the real world experiments for that reason.
> >
> > >Lack of multi-task manipulation results in real robot experiments. Both PerAct [1] and GNFactor [2] have shown ability to execute real-world multi-task manipulation, and it could be good to compare the multi-task performance in real robot also, considering this work is closely related to PerAct [1] and GNFactor [2].
> >
> > You are absolutely correct, we did not have the time to do that, it is within the capabilities of our model. We leave this for future work.
> >
> > ### 5. Comparison to GNFactor
> > >Lack of discussion and experiment comparison with recent related works such as GNFactor [2]. I think this very recent method [1] could possibly serve as a baseline and it would be good to see some direct experiment results.
> >
> > Our experiments follow the established train and test splits and multi-task experimental setup first introduced by PerAct, which considers 18 manipulation tasks, 100 training demonstrations per task and 4 cameras per task. Our model and baselines are all trained and tested in the same setup.
> >
> > GNFactor focuses on 20 demonstrations single-camera setup and uses only 10 out of the 18 tasks.
> > Thus, this does not allow any comparison with related works.
> > Moreover, the four-camera **10-demo 18-task** PerAct model has much higher performance  in the PerAct paper than what is reported in the GNFactor paper by the  four-camera **20-demo 10-task** Per-Act baseline ( 46.4% average performance across 10 tasks, versus 22.7% reported in the GNFactor paper). Due to these discrepancies, we did not know how to judge their setup.
> >
> > Following your suggestion, we will report our performance with one camera and 20 demonstrations on these 10 tasks to be able to compare with GNFactor. Thank you for your suggestion.
> >
> >
> > ### 6. Typo in Figure 2 (a): Acter -> Actor
> > Thank you for catching this, we fixed it.

---

> ### Comment · Reviewer_sG3K · 2023-11-21
> **Thank the authors for response**
>
> Thank the authors for detailed explanations. I have the following further questions.
>
> ## 4. Real-robot experiments.
> > This is hard to do as it requires resetting the real environment to identical state across multiple models. PerAct, Act3D, RVT, InstructRL and most other works do not consider baselines in the real world experiments for that reason. You are absolutely correct, we did not have the time to do that, it is within the capabilities of our model. We leave this for future work.
>
> Considering the time given in the discussion period (~12 days), I do not think any initial real robot experiments (e.g., one task and 10 trials) could not be conducted. Training one baseline agent might take mostly 1-2 days.
>
> And yes, you are right that most of the papers you mention do not compare other baselines in the real world. However, as the authors describe, the setting is closely following PerAct [1]. PerAct is not comparing with other baselines mainly because there are no competitive baselines. In addition, there are also papers in a close setting that actually do, e.g., GNFactor [2].
>
> Overall, I do not think the excuse from the authors holds.
>
>
> ## 5 Comparision to GNFactor
> > Following your suggestion, we will report our performance with one camera and 20 demonstrations on these 10 tasks to be able to compare with GNFactor. Thank you for your suggestion.
>
> It seems the updated paper does not include any new discussion or new experiments regarding this question.
>
> > Due to these discrepancies, we did not know how to judge their setup.
>
> Again, this discrepancy can not hold as an excuse for no discussion and no experiments, since the code of GNFactor [2] is available online, and the small discrepancy (only the task and the number of demonstrations) is very common for imitation learning papers. Comparing them under the same setting is not an impossible thing, let alone with enough time for the discussion period.
>
>
> I would maintain my score, but be willing to raise my score if the problems are addressed better.
>
> [1] Shridhar, Mohit, et al. "Perceiver-actor: A multi-task transformer for robotic manipulation." CoRL, 2022.
>
> [2] Ze, Yanjie, et al. "Gnfactor: Multi-task real robot learning with generalizable neural feature fields." CoRL, 2023.

---

### Official Review · Reviewer_Ws6y · 2023-10-31

**Soundness:** 3 good
**Presentation:** 2 fair
**Contribution:** 2 fair
**Rating:** 1
**Confidence:** 4

**Summary:**

This paper introduces a framework that marries 3D scene representations and diffusion policies for imitation learning of robot manipulation tasks. The proposed scene representation is a 3D feature cloud fused from multi-view, multi-scale CLIP features using depth maps, which encodes semantics and spatial information. The diffusion policy captures the multimodality of action distribution. In the experiments, the proposed method is compared against multiple baselines on the simulated RLBench dataset, and tested on 5 real-world tasks. Ablation studies further verify the necessity of the 3D representation and the use of relational attention.

Note that many of the design choices and setups in this work are largely based on Act3D [1].

[1] Theophile Gervet, Zhou Xian, Nikolaos Gkanatsios, and Katerina Fragkiadaki. Act3d: Infinite
resolution action detection transformer for robotic manipulation. CORL 2023.

**Strengths:**

This paper is probably the first work to combine 3D scene representations with diffusion policy for learning robot manipulation tasks. The proposed framework demonstrates good performance in both simulation and real-world experiments, and establishes a new SOTA on RLBench tasks.

**Weaknesses:**

My initial impression of this paper is that some aspects of the method lack clarity, and certain paragraphs appear somewhat inconsistent. I found myself confused about specific technical details until I reviewed Act3D [1], a previous paper that this work heavily draws upon.

**My primary concern about this work is that a portion of the technical method and writting appears to be directly borrowed from [1]. However, this relationship is not transparently acknowledged.**

1. The proposed framework adopts the 3D scene representation (in a simplified form), 3D relational transformer, and training/evaluation setups from [1], which is never explicitly mentioned in the paper.
2. The main contribution of this work lies in the use of a diffusion policy to capture multimodal action distributions. However, this aspect is not extensively discussed or thoroughly evaluated.
3. Not only the writing style of this paper closely resemble Act3D [1], but some paragraphs have similar or even same counterparts in [1]. Detailed in ethics review.

**Questions:**

N/A

**Details Of Ethics Concerns:**

Many paragraphs are directly borrowed from a previous work [1], some with minor rephrase. This potentially forms plagiarism.

| level of similarity | paragraph in this paper | counterpart in [1]  |
|----------|----------|----------|
| Rephrased  |  “2D and 3D scene representations for robot manipulation” in Section 2   | "Learning robot manipulation from demonstrations" in Section 2   |
| Rephrased & adapted  | the first two paragraphs in Section 3.2   | Section 3   |
| Rephrased (almost the same)  | "Scene and language encoder" in Section 3.2 | "Visual and language encoder" in Section 3 |
| Partially the same | "Baselines" and "Evaluation metrics" in Section 4 | "Baselines" and "Evaluation metric" in Section 4.1 |

[1] Theophile Gervet, Zhou Xian, Nikolaos Gkanatsios, and Katerina Fragkiadaki. Act3d: Infinite
resolution action detection transformer for robotic manipulation. CORL 2023.

---

> ### Author Response · Authors · 2023-11-15
> **Authors' Response**
>
> ### 1. Relationship to Act3D
>
> >My primary concern about this work is that a portion of the technical method and writting appears to be directly borrowed from [1]. However, this relationship is not transparently acknowledged.
>
>
> **Please, see the general response in the beginning of the rebuttal.**
>
>
> > The proposed framework adopts the 3D scene representation (in a simplified form), 3D relational transformer, and training/evaluation setups from [1], which is never explicitly mentioned in the paper.
>
>
> **Please, see the general response in the beginning of the rebuttal.**
>
>
>
>
>
> ### 2. Discussion on multimodality and contribution
> >The main contribution of this work lies in the use of a diffusion policy to capture multimodal action distributions. However, this aspect is not extensively discussed or thoroughly evaluated.
>
>
>
> Motivated by your comment and the comment of Reviewer sG3K, and in an attempt to validate the argument regarding utility for diffusion, we conducted an additional experiment where we use a close to **identical** architecture to 3D Diffuser Actor for **regressing** the 3D location and 3D rotation of the target keypose, as opposed to the noise and the use of iterative denoising (no denoising iteration index input for  regression). We show the comparison for 3D location prediction on train and val sets ([here](https://docs.google.com/presentation/d/e/2PACX-1vTZkgzAxzw2IlVY86Ft9AmRC2UR_RBVFV8uFBdeyxeg9hdCWN7mstblmIkOQAogGR2LT1k3dsLEXcTo/pub?start=false&loop=false&delayms=3000)) and the comparison for 3D rotation prediction on train and val sets ([here](https://docs.google.com/presentation/d/e/2PACX-1vSkoF10a1x7D12Ll1xCeJ4rPSA02GzTKPsCq0yS8dJWsZYuUnFAGHjCixD_blR1_hyDBdluIqTtpbsF/pub?start=false&loop=false&delayms=3000#slide=id.g29b18ed0d7c_0_0)).
>
> As you can see, 3D diffuser actor generalizes much better than the regression model, especially for 3D location.  Both models use the same scene encoder, language encoder, and 3D relative transformer decoder architecture, but one predicts the end-effector through iterative denoising, while the other(regression) through a single feedforward pass.
>
> **Our contribution**: We propose the first 3D keypose diffusion model denoised from 3D scene representations for learning 3D manipulation from demonstrations. We achieve superior performance over many prior methods in apples-to-apples comparisons in an established experimental setup on RLBench(+12% compared to state-of-the-art methods).   Our model, 3D Diffuser Actor, marries 3D representations with diffusion policies for robot manipulation, as  we describe  clearly in the abstract (*"a framework that marries diffusion policies and 3D scene representations for robot manipulation"*) and in the intro. In both abstract and intro  we first describe diffusion policies (in abstract: *"Diffusion policies capture the action distribution conditioned on the robot and environment state using conditional diffusion models. They have recently shown to outperform both deterministic and alternative generative policy formulations in learning from demonstrations."*), then 3D policies (in abstract: *"3D robot policies use 3D scene feature representations aggregated from single or multiple 2D image views using sensed depth. They typically generalize better than their 2D counterparts in novel viewpoints."*) and we say we introduce 3D diffusion policies that marry the above two: (in abstract: *"We unify these two lines of work and present a neural policy architecture that uses 3D scene representations to iteratively denoise robot 3D rotations and translations for a given language task description."*)
> **The specific way this marriage is accomplished is important**:  we represent the current noisy end-effector pose estimate as a 3D token in the 3D feature cloud and featurize it  with relative 3D attentions. Since the ICLR submission deadline, **we ran the exact same model with absolute position attention (closer to what 2D diffusion policies do) on the whole PerAct benchmark and obtained an 8% worse performance**. We consider the above 3D diffusion keypose policy model to be our contribution.
>
>
> **Please, also see the general response in the beginning of the rebuttal.**

---

> > ### Comment · Reviewer_Ws6y · 2023-11-22
> >
> > I would like to thank the authors for their time and efforts to prepare the rebuttal and revise the paper.
> >
> > The paper looks less similar to Act3D after the authors rewrite some of the paragraphs and explicitly acknowledge borrowing technical contents from it. However, as the scene representation it uses can be trivially adapted from Act3D, replacing the regression policy head in Act3D with a diffusion model is an incremental contribution.
> >
> > The extra experimental results reveal that the diffusion head generalizes better than the regression head, which makes sense to me. But as a major claim of the paper is "handling multi-modality", it will be nice to explicitly demonstrate the multi-modality robot behavior qualitatively or quantitatively. A good example can be found in Figure 4 of the diffusion policy paper [1].
> >
> > [1] Chi, Cheng, et al. "Diffusion policy: Visuomotor policy learning via action diffusion." RSS 2023.
> >
> > I will keep my rating as it is.

---

### Official Review · Reviewer_7nbv · 2023-10-31

**Soundness:** 4 excellent
**Presentation:** 2 fair
**Contribution:** 3 good
**Rating:** 6
**Confidence:** 4

**Summary:**

This paper proposes 3D Diffuser Actor, a transformer-based behavior-cloning method that combines the power of diffusion policies and 3D scene representations. The model tokenizes multi-view camera observations, language instructions, and the history proprioception information. A 3D relative transformer models the diffusion process, processing these tokens to predict the next gripper pose over several diffusion steps. The experimental results showcase the remarkable performance of the proposed method, outperforming several strong baselines in the RLBench environments.

**Strengths:**

The experimental results are exceptionally strong, demonstrating a substantial improvement over various recently proposed baselines (from late 2022 to mid-2023).

**Weaknesses:**

1. The method's description lacks clarity, particularly concerning crucial components of the architecture. The appendix does not sufficiently clarify these ambiguities either.
2. The backbone of the proposed network is extremely similar to Act3D, including using multi-view image input, using the pyramid network for feature exaction, the generation of the 3D feature cloud, and the 3D relative transformer. I understand that Act3D is a very recent work, however, as the authors are already aware of Act3D, proper discussion regarding the relationship between this work and Act3D should be addressed.

**Questions:**

1. When building the 3D scene feature cloud, the authors claim, `We associate every 2D feature grid location in the 2D feature maps with a depth value, by averaging the depth values of the image pixels that correspond to it.` What is a `grid` here? Is each 2D feature map from the feature pyramid network separated into an NxN grid akin to ViT?
2. When generating the 3D feature cloud, how does the method handle cases where multiple points (pixels) from different views correspond to the same 3D location? Are the features averaged in such instances?
3. What constitutes a visual token in this context? Are individual 3D points considered tokens?
4. The performance of the proposed method is notably poor in close jar and sort shape. Could the authors provide a detailed failure analysis to shed light on these issues?
5. Given that diffusion policies suggest the utility of diffusing multiple action steps into the future, has the method been evaluated with multiple keypoint steps in the diffusion process?

---

> ### Author Response · Authors · 2023-11-15
> **Authors' Response**
>
> ### 1. Same backbone as Act3D
> > The backbone of the proposed network is extremely similar to Act3D, including using multi-view image input, using the pyramid network for feature exaction, the generation of the 3D feature cloud, and the 3D relative transformer. I understand that Act3D is a very recent work, however, as the authors are already aware of Act3D, proper discussion regarding the relationship between this work and Act3D should be addressed.
>
> **Please, see  the general response in the beginning of the rebuttal regarding the relationship between this work and Act3D.**
>
> ### 2. Explanation of 3D feature cloud
> > When building the 3D scene feature cloud, the authors claim, We associate every 2D feature grid location in the 2D feature maps with a depth value, by averaging the depth values of the image pixels that correspond to it. What is a grid here? Is each 2D feature map from the feature pyramid network separated into an NxN grid akin to ViT?
>
> An image encoder extracts a $H \times W \times C$ feature map from each camera view, where $H$, $W$, and $C$ denotes the height, width and channel dimension of the feature map.  Each $1 \times 1 \times C$ feature vector (we call it token) in the feature map corresponds to a fixed-size patch in the input image. We can thus lift a 2D feature token to 3D by averaging the 3D locations of all pixels within the patch. The 3D locations of pixels are obtained by combining the 2D pixel locations and depth values (x, y, d) using the camera intrinsics and pinhole camera equation. We updated the manuscript and Figure 1 to explain the feature extraction and 2D-3D lifting procedure more clearly.
>
> > When generating the 3D feature cloud, how does the method handle cases where multiple points (pixels) from different views correspond to the same 3D location? Are the features averaged in such instances?
>
> We found that having two points with identical 3D locations rarely happens, so we did not explicitly handle this case. Although some merging could be applied, such as averaging, we adopted a more general solution and simply let attention focus on the important features.
>
> >What constitutes a visual token in this context? Are individual 3D points considered tokens?
>
> Each  feature vector from the 2D feature maps lifted to 3D is a 3D token.
>
>
>
> ### 3. Predicting further in the future
> >Given that diffusion policies suggest the utility of diffusing multiple action steps into the future, has the method been evaluated with multiple keypoint steps in the diffusion process?
>
> Thank you for your suggestion. Although in this version we only predict the next keypose, our model could indeed support the prediction of multiple keyposes and it would be interesting to explore that in the future as a way to plan. Nonetheless, we would like to emphasize that our diffusion model predicts the next subgoal to reach, not low-level actions directly. Instead, once a subgoal keypose has been predicted, then a low-level planner is assigned with predicting a sequence of actions that connect the current pose and the predicted keypose. This decomposition has been followed by previous approaches as well (e.g. C2F-QA, PerAct, Act3D). This is different from what related works on diffusion policies do, which is to predict low-level actions directly.

---

### Official Review · Reviewer_2s3N · 2023-11-03

**Soundness:** 2 fair
**Presentation:** 3 good
**Contribution:** 2 fair
**Rating:** 3
**Confidence:** 2

**Summary:**

This paper presents a framework called "3D Diffuser Actor," combining diffusion policies and 3D scene representations to enhance robot manipulation. Along with 3D scene features aggregated from single or multiple 2D image views using sensed depth, these policies enable improved generalization over 2D counterparts. The new model’s architecture uses 3D scene representations to iteratively rectify robot 3D rotations and translations given a language description of a task. The experiments demonstrate the efficacy of the proposed framework by outperforming previous benchmarks in learning from demonstrations, both in simulated environment and real world.

**Strengths:**

The authors conduct a thorough evaluation of their method as they evaluate their method in both simulated environment and real world, and comparing them with existing strong baselines. A thorough evaluation have us better understand the proposed model and their actual performance.

**Weaknesses:**

1. The scientific contribution of this paper is unclear. Diffusion model could not be the contribution of this paper, and adopting diffusion mode for trajectory generation is not a new idea (see [a]).
2. The figures in this paper lack sufficient illustrative value and information. For instance, Figure 1 appears to show the model taking a multi-view image as input, yet the caption indicates it uses a 3D scene feature cloud. A clear connection between these two elements would greatly improve understanding.

**Questions:**

Your diffusion model appears to produce only the target pose of the action, after which a trajectory is generated using the MoveIt planner given the initial joint state and target joint state. Does the robot execute the entire trajectory directly until the end, or does it update the target pose using the diffusion model and regenerate the trajectory after each forward step?

---

> ### Author Response · Authors · 2023-11-15
> **Authors' Response**
>
> ### 1. Contribution
> > The scientific contribution of this paper is unclear. Diffusion model could not be the contribution of this paper, and adopting a diffusion model for trajectory generation is not a new idea (see [a]).
>
>
>
>
> **Our contribution**: We propose the first 3D keypose diffusion model denoised from 3D scene representations for learning 3D manipulation from demonstrations. We achieve superior performance over many prior methods in apples-to-apples comparisons in an established experimental setup on RLBench(+12% compared to state-of-the-art methods).   Our model, 3D Diffuser Actor, marries 3D representations with diffusion policies for robot manipulation, as  we describe  clearly in the abstract (*"a framework that marries diffusion policies and 3D scene representations for robot manipulation"*) and in the intro. In both abstract and intro  we first describe diffusion policies (in abstract: *"Diffusion policies capture the action distribution conditioned on the robot and environment state using conditional diffusion models. They have recently shown to outperform both deterministic and alternative generative policy formulations in learning from demonstrations."*), then 3D policies (in abstract: *"3D robot policies use 3D scene feature representations aggregated from single or multiple 2D image views using sensed depth. They typically generalize better than their 2D counterparts in novel viewpoints."*) and we say we introduce 3D diffusion policies that marry the above two: (in abstract: *"We unify these two lines of work and present a neural policy architecture that uses 3D scene representations to iteratively denoise robot 3D rotations and translations for a given language task description."*)
> **The specific way this marriage is accomplished is important**:  we represent the current noisy end-effector pose estimate as a 3D token in the 3D feature cloud and featurize it  with relative 3D attentions. Since the ICLR submission deadline, **we ran the exact same model with absolute position attention (closer to what 2D diffusion policies do) on the whole PerAct benchmark and obtained an 8% worse performance**. We consider the above 3D diffusion keypose policy model to be our contribution.
> **Please, also see the general response in the beginning of the rebuttal.**
>
>
> ### 2. Explanation of 3D feature cloud
> > The figures in this paper lack sufficient illustrative value and information. For instance, Figure 1 appears to show the model taking a multi-view image as input, yet the caption indicates it uses a 3D scene feature cloud. A clear connection between these two elements would greatly improve understanding.
>
> An image encoder extracts a $H \times W \times C$ feature map from each camera view, where $H$, $W$, and $C$ denotes the height, width and channel dimension of the feature map.  Each $1 \times 1 \times C$ feature token in the feature map corresponds to a fixed-size patch in the input image. We can thus lift a 2D feature token to 3D by averaging the 3D locations of all pixels within the patch. The 3D locations of pixels are obtained by combining the 2D pixel locations and depth values (x, y, d) using the camera intrinsics and pinhole camera equation. We updated the manuscript to explain the feature extraction and 2D-3D lifting procedure more clearly. We also **updated Figure 1** to illustrate the lifting process.
>
> ### 3. Frequency of keypose prediction
> > Your diffusion model appears to produce only the target pose of the action, after which a trajectory is generated using the MoveIt planner given the initial joint state and target joint state. Does the robot execute the entire trajectory directly until the end, or does it update the target pose using the diffusion model and regenerate the trajectory after each forward step?
>
> Once a keypose is predicted, a low-level planner is tasked to predict a trajectory between the current and the predicted pose. This trajectory is executed open-loop until the end. We found that this was sufficient for all 18 tasks we focus on. At the same time, it’s more efficient computationally, since the denoising process is sparsely called.

---

> ### Comment · Reviewer_2s3N · 2023-11-23
>
> I would like to thank the authors for their comprehensive responses, touching up of the figures.
>
> The overall writing of this paper is more clear. However, after going through all the comments (from the authors and other reviewers), I am still not convinced by the claimed contributions. The proposed method is more of an assemble of existing techniques without providing adequate sharp insight the of manipulation problem.
>
> I will downgrade my rating as the authors does not provide adequate strong argument for their contributions and evidence for the advantages of the proposed method in real world.
>
> Plus, I also change my rating of the presentation of this paper from 2 to 3.

---

### Author Response · Authors · 2023-11-15
**General Response**

Thank you for your comments and for  reading our paper and the Act3D baseline. In this joint response we explain the relations between Act3D, 3D Diffuser Actor and ChainedDiffuser. We then provide individual responses to each reviewer's concerns.

### Relation of 3D Diffuser Actor to Act3D
**Act3D** is a method that learns to predict 3D keyposes of the robot's end-effector conditioned on  multiview RGB-D images and a language instruction using learning from demonstrations. Its goal is to **predict a 3D action map** so that it preserves equivariance which helps generalization in action prediction.
**Main contribution**: A high resolution 3D action map typically requires a high resolution 3D voxel grid to featurize. This is a problem since it is prohibitively expensive to run all-to-all attentions across all 3D voxels.   Act3D foregoes 3D voxelization altogether and instead uses  a coarse-to-fine 3D point sampling and featurization, to build 3D action maps of varying 3D spatial resolution (we say this in our related work: *"Recently, Act3D (Gervet et al., 2023) uses coarse-to-fine 3D attentions in a 3D scene feature cloud and avoids 3D voxelization altogether. It samples 3D points in the empty workspace and featurizes them using cross-attentions to the physical 3D point features."*).
**Details**: Act3D instantiates a  scene 3D feature cloud by featurizing   multiview RGB  images and lifting the patch features to 3D using sensed depth. It also encodes the language instruction using CLIP's language encoder. It then samples 3D  points (they call them ghost points) on a 3D grid and featurize them by cross-attending to the scene 3D feature cloud. The scene 3D feature cloud is never updated. A learnable query also cross-attends to the scene feature cloud  and it is used to score all 3D ghost points and predict a 3D action map in this way. This process is repeated 3 times, each time with a refined 3D ghost point sampling and more fine-grained visual features to improve map resolution. At the end, the query features are also used to regress to a 3D rotation of the end-effector.



**3D Diffuser Actor** is a method that learns to predict 3D keyposes of the robot's end-effector conditioned on  multiview RGB-D images and a language instruction using learning from demonstrations  (this is identical to Act3D).  **It foregoes predicting 3D action maps altogether. Its bet is that the iterative denoising process will be able to  handle multimodality in 3D end-effector location as well as or better than 3D action maps, while in addition handling multimodality in 3D rotation (in Act3d rotation is simply regressed). The hope is that these two will lead to better generalization in 3D keypose prediction**
**Main contribution**: 3D Diffuser Actor casts 3D keypose prediction as a  denoising  process conditioned on multiview images and language instructions. It proposes a denoising network that takes as input multiview images, a language instruction, **a current estimate of the end effector’s 3D rotation and 3D translation and the index of the denoising iteration of diffusion, and outputs the noise in the 3D translation and rotation of the end effector.** It featurizes  the RGB multiview images using a 2D pretrained backbone (CLIP) and lifts the 2D feature maps  into a 3D featured cloud. It then embeds the language instruction using a CLIP language encoder (similar to Act3D, CLIPort (Shridhar et al., CoRL 2021) among others), it uses FiLM layers to input the denoising iteration and instantiates a 3D token for the current end-effector estimate. It uses relative 3D position attentions to fuse all this information (as we say we are inspired by Act3D at the top of page 6). 3D Diffuser's Actor  3D relative transformer is very different from Act3D's: **3D Diffuser Actor does not use 3D ghost point grids or coarse-to-fine featurizations**. The noisy keypose token, visual tokens and proprioception tokens are fed to a 3D relative self-attention layer that also updates the scene feature tokens. We refer to our Figure 6 in the Appendix for an illustration of our denoising 3D transformer model.

---

> ### Author Response · Authors · 2023-11-15
> **General Response (2)**
>
> The models of Act3D and 3D Diffuser Actor are thus  different in that:
> 1. One is a diffusion model the other is a deterministic model
> 2. They have different inputs and outputs as noted above. One maps x to y directly through coarse to fine 3D point sampling for 3D location and regression for 3D rotation, the other is a denoising diffusion model that takes x and the current y estimate (3D location and 3D rotation) and iteration index and predicts the noise in the current estimate.
> 3. There is no coarse-to-fine sampling or any form of point sampling in 3D Diffuser Actor.  3D Diffuser Actor does not build a 3D map (a map of 3D locations where each one is scored whether it contains the next end-effector 3D keypose or not).
> 4. The 3D relative transformers differ as noted above: 1) the token sets are very different as explained above, 2) one is a cross-attention and the other a self-attention.
>
>
> The models of Act3D and 3D Diffuser Actor are similar in that:
>
> 1. They both predict 3D keyposes for the end-effector and are trained from demonstations. Many more methods live in this setting such as: PerAct, RVT, InstructRl, HiveFormer, C2F-QA.
> 2. They lift RGB-D images into 3D feature clouds and use 3D transformers with relative attentions (a similarity we acknowledge at the top of page 6 and clearly explain in the related work).
>
>
> We did not acknowledge the 3D feature building because we believe it is a standard way of obtaining a 3D feature point cloud. For example, it has been used before in:
> 1. OpenScene: 3D Scene Understanding with Open Vocabularies, https://openaccess.thecvf.com/content/CVPR2023/papers/Peng_OpenScene_3D_Scene_Understanding_With_Open_Vocabularies_CVPR_2023_paper.pdf
> 2. Virtual Multi-view Fusion for 3D Semantic Segmentation https://arxiv.org/abs/2007.13138.
> 3. ChainedDiffuser(discussed later)
>
>
> **We cited Act3D 10 times in our submitted paper, and it is our most competitive baseline. We clearly do not claim any of its contributions. In the submitted paper we do not claim 3D representations to be our contribution**.
>
>
> ### Our contribution
>  We propose the first 3D keypose diffusion model denoised from 3D scene representations for learning 3D manipulation from demonstrations. We achieve superior performance over many prior methods in apples-to-apples comparisons in an established experimental setup on RLBench(+12% compared to state-of-the-art methods).   Our model, 3D Diffuser Actor, marries 3D representations with diffusion policies for robot manipulation, as  we describe  clearly in the abstract (*"a framework that marries diffusion policies and 3D scene representations for robot manipulation"*) and in the intro. In both abstract and intro  we first describe diffusion policies (in abstract: *"Diffusion policies capture the action distribution conditioned on the robot and environment state using conditional diffusion models. They have recently shown to outperform both deterministic and alternative generative policy formulations in learning from demonstrations."*), then 3D policies (in abstract: *"3D robot policies use 3D scene feature representations aggregated from single or multiple 2D image views using sensed depth. They typically generalize better than their 2D counterparts in novel viewpoints."*) and we say we introduce 3D diffusion policies that marry the above two: (in abstract: *"We unify these two lines of work and present a neural policy architecture that uses 3D scene representations to iteratively denoise robot 3D rotations and translations for a given language task description."*)
> **The specific way this marriage is accomplished is important**:  we represent the current noisy end-effector pose estimate as a 3D token in the 3D feature cloud and featurize it  with relative 3D attentions. Since the ICLR submission deadline, **we ran the exact same model with absolute position attention (closer to what 2D diffusion policies do) on the whole PerAct benchmark and obtained an 8% worse performance**. We consider the above 3D diffusion keypose policy model to be our contribution.

---

> > ### Author Response · Authors · 2023-11-15
> > **General Response (3)**
> >
> > In the following, we address specific concerns raised by  Reviewer Ws6y regarding similarity of passages in our paper with Act3D:
> >
> > **Related work paragraph**
> > When  two works are competitors for the same problem domain and for the same benchmark and are submitted very close in time to one another, it might be natural to have similar content in the related work.  We have re-worded this paragraph to  make it less similar.
> >
> >
> > **"Scene and language encoder" in Section 3.2**:
> > Our 3D representation builds upon Act3D: we encode each RGB image with a pre-trained 2D  backbone (CLIP) and lift the  feature vectors of the 2D feature maps to 3D using  sensed depth.  **This is a standard way of obtaining a 3D feature point cloud**, e.g., similar has been used in
> > 1. OpenScene: 3D Scene Understanding with Open Vocabularies, https://openaccess.thecvf.com/content/CVPR2023/papers/Peng_OpenScene_3D_Scene_Understanding_With_Open_Vocabularies_CVPR_2023_paper.pdf and
> > 2. Virtual Multi-view Fusion for 3D Semantic Segmentation https://arxiv.org/abs/2007.13138.
> >
> > Since submission we found a multiscale 3D feature cloud and pyramic network not to be important in performance.
> >
> > Our language instruction is encoded using a CLIP langage encoder **which is standard and has been used in Act3D, CLIPort, PerAct,  and many many others**.
> > We use relative 3D attentions which was **a novel characteristic of Act3D and thus we mention at the top of page 6 “Inspired by recent work in visual correspondence (Li & Harada, 2022; Gkanatsios et al., 2023b) and 3D manipulation (Gervet et al., 2023-”Act3D-”), we use rotary positional embeddings.“**  You are absolutely correct that very similar passages across papers is not supposed to happen, despite the fact it is common practise for papers that build upon one another. We added the following sentence in the beginning of our Scene and language encoder paragraph to make transparent the similarity to Act3D:  *“We adopt a scene and language encoder similar  to Act3D \cite{gervet2023act3d}. We describe it here to make the paper self-contained.”*.
> >
> > **The first two paragraphs in Section 3.2**: We use the same setup as PerAct and Act3D regarding predicting 3D **keyposes** as opposed to continuous end-effector **trajectories**. Each keypose is  represented by a 3D rotation and 3D translation. This setup was not invented by Act3D, but rather inherited from previous works (C2F-QA, PerAct, HiveFormer etc.), which was in turn inherited by the Q-attention work of James and Davison that we cite.
> > We represent rotation differently than Act3D and we write: *"We represent rotations using the 6D rotation representation of Zhou et al. (2020) for all environments in all our experiments, to avoid the discontinuities of the quaternion representation that previous approaches use (Guhur et al., 2023; Gervet et al., 2023)."* We have added the following sentence in the beginning of our method section: *“We adopt the same keypose prediction framework from PerAct and Act3D, which we describe below for completeness.”*
> >
> > **"Baselines" and "Evaluation metrics" in Section 4** The evaluation metrics used are standard in all 3D manipulation papers. 2 out of 6 baselines are the same. We share instructRL and PerAct as baselines.	We have added the following sentence in the beginning of our Evaluation metrics section: *“Following previous work \cite{gervet2023act3d,shridhar2023perceiver},...”

---

> > > ### Author Response · Authors · 2023-11-15
> > > **General Response (4)**
> > >
> > > ### Relation of 3D Diffuser Actor to ChainedDiffuser
> > > **ChainedDiffuser** is a  method that **takes as input 3D keyposes** predicted by an off-the-shelf  3D keypose prediction method, such as PerAct or Act3D, and **predicts end-effector trajectories to link the current keypose to the predicted one.**
> > > **Main contribution**: ChainedDiffuser proposes to get rid of sampling-based motion planners (like RRT) because they fail in tasks that cannot be easily decomposed into straight-line trajectory segments, such as opening doors, wiping the table, etc… and instead use trajectory diffusion that conditions on the current and predicted  3D end-effector keyposes  to predict a 3D end-effector trajectory that links them.
> > >
> > > **Details**: ChainedDiffuser casts current to predicted 3D keypose motion planning as trajectory  diffusion. It uses an architecture similar to Act3D to create a 3D feature cloud and takes as input the current trajectory estimate to predict scaled noise for translation trajectory and the final denoised rotation trajectory. It shows  that **in tasks that require continuous motion**, Act3D keyposes in combination with this trajectory denoising from keypose to keypose, much outperforms Act3D+motion planners.  ChainedDiffuser’s contribution is a learnt trajectory predictor that works alongside an off-the-shelf 3D keypose prediction method. **ChainedDiffuser is complementary to our work: We can use keyposes predicted by 3D Diffuser Actor and link them with ChainedDiffuser’s trajectories  instead of motion planners and achieve similar boosts in these tasks**. Because the tasks in the PerAct benchmark that we consider do not require such continuous trajectories, we did not consider it for comparison, because in these specific tasks we consider motion planners work well; ChainedDiffuser performs as well as Act3D there (no additional performance boost).
> > >
> > >
> > > The models of ChainedDiffuser and 3D Diffuser Actor are thus  different in that:
> > > **1. ChainedDiffuser takes as input the to-reach 3D keypose while 3D Diffuser Actor tries to predict it.** In that sense, 3D Diffuser Actor has a much much harder task to solve.
> > > 2. 3D Diffuser Actor denoises 3D keyposes and ChainedDiffuser denoises 3D trajectories.
> > > 3. The experiments in ChainedDiffuser concern single manipulation tasks while 3D Diffuser Actor shows multi-task results across the whole PerAct suite.
> > >
> > > The models of ChainedDiffuser and 3D Diffuser Actor are   similar in that:
> > > 1. Both build 3D feature clouds and use diffusion to predict their desired output, being the next 3D keypose for 3D Diffuser Actor and being a current to predicted 3D keypose  trajectory for ChainedDiffuser.
> > >
> > > ChainedDiffuser can take as input the keyposes predicted by 3D Diffuser Actor and we explained above why we did not consider this.
> > >
> > >
> > > Though we were aware of ChainedDiffuser during submission of 3D Diffuser Actor, we did not cite it because it was not on arxiv yet nor on openreview with camera ready. We could have cited it as anonymous using the open review bibtex, as we see now.  It is our mistake that we did not cite it,  because it is a diffusion model in robotics and also uses 3D representations. We have updated the manuscript with orange in the related work citing the chained-diffuser paper.
> > >
> > >
> > >
> > > Thank you very much for your comments which push us to much more clearly describe our work and its relations to   recent approaches, as well as fix numerous typos in the submitted paper. Please, do not hesitate to reply back if you need any more information for making your decisions.

---

> ### Author Response · Authors · 2023-11-23
> **General Response (5)**
>
> ### Low-performance on the `close_jar` task
>
> Thank you for pointing this out. After looking into this issue, we found that the success conditions of the task might not be correct.  We measured the success rate of the expert demonstrations on the validation set, where [demonstrations](https://github.com/peract/peract#pre-generated-datasets) and [RLBench library](https://github.com/MohitShridhar/RLBench/tree/peract) are released by PerAct.  Surprisingly, we found that all ground-truth demonstrations fail: the overall success rate is 0% by replaying demonstrations.  We contacted Act3D authors, who confirmed our findings and their modification of the success condition. We will report this issue to the PerAct authors so the benchmark can be corrected. Using the fixed success conditions from Act3D, **3D Diffuser Actor achieves 97\% on close_jar**. Thank you for pushing us in this investigation as we believe this will correct a benchmark that will be used by many researchers.

---

> ### Author Response · Authors · 2023-11-23
> **General Response (6)**
>
> ### Additional experiments: Complete ablations on 18 RLBench tasks
>
> Since submission, we were able to expanded our ablations to the full  PerAct 18-task benchmark (in the submitted paper we only show ablations across indivisual tasks due to lack of resources).
> Specifically, we consider the following ablative versions of our model which we introduced in our  submission and we present again here for completeness:
>
> * *2D Diffuser Actor*, is a method identical to 3D Diffuser Actor but uses 2D scene representations similar to (Chi et al., 2023), instead of 3D. We remove the 3D scene encoding from 3D Diffuser Actor, and instead generate per-image 2D representations by average-pooling features within each view, and add learnable embeddings to distinguish different views, following (Chi et al., 2023). We use  attention layers for jointly encoding the action estimate and 2D image features, following (Chi et al., 2023)
>
> * *3D Diffuser Actor w/o RelAtt*, an ablative version of our model that uses standard non-relative attentions to featurize the current rotation and translation estimate with the 3D scene feature cloud, and does not ground the current gripper estimate in the scene.
>
> As shown in the following table, our 3D Diffuser Actor consistently outperforms these two ablative versions among most tasks. The results verifies the two main claims of this work:
> *  explicitly using 3D scene representations over 2D ones is crucial for learning to manipulate and
> *  representing the gripper as a scene token and contextualizing it with relative attentions is important for generalization. We will include the new results in the final version of the paper.
>
>
> \begin{array}{l|c|c|c|c|c|c|c|c|c|c}
>  & \small\text{Avg.} & \small\text{open} & \small\text{slide} & \small\text{sweep to} & \small\text{meat off} & \small\text{turn} & \small\text{put in} & \small\text{drag} & \small\text{place}& \small\text{close}  \newline
>  & \small\text{Success} & \small\text{drawer} & \small\text{block} & \small\text{dustpan} & \small\text{grill} & \small\text{tap} & \small\text{drawer} & \small\text{stick} & \small\text{cups} & \small\text{jar}  \newline \hline
> \small\textbf{PerAct} & \small43 & \small80 & \small72 & \small56 & \small84 & \small80 & \small68 & \small68 & \small0 & \small60 \newline
> \small\textbf{RVT} & \small63 & \small71 & \small82 & \small72 & \small88 & \small94 & \small88 & \small\textbf{99} & \small4 & \small52 \newline
> \small\textbf{Act3D} & \small65 & \small93 & \small93 & \small92 & \small94 & \small94 & \small90 & \small92 & \small3 & \small92 \newline \hline
> \small\textbf{2D Diffuser Actor} & \small44 & \small68 & \small65 & \small90 & \small75 & \small72 & \small55 & \small79 & \small0 & \small94 \newline
> \small\textbf{3D Diffuser Actor without Rel. Attn.} & \small68 & \small96 & \small\textbf{99} & \small87 & \small\textbf{98} & \small\textbf{97} & \small91  &\small 98  & \small0 & \small94 \newline
> \small\textbf{3D Diffuser Actor (ours)} & \small\textbf{79} \small\textbf{} & \small87 & \small97 & \small\textbf{99} & \small95 & \small\textbf{97} & \small\textbf{92} & \small\textbf{100}  & \small\textbf{32} & \small\textbf{97} \newline
> \hline
>  & \small\text{stack} & \small\text{screw} & \small\text{put in} & \small\text{place} & \small\text{put in} & \small\text{sort} & \small\text{push} & \small\text{insert} & \small\text{stack} \newline
>  & \small\text{blocks}  & \small\text{bulb} & \small\text{safe} & \small\text{wine} & \small\text{cupboard} & \small\text{shape} & \small\text{buttons} & \small\text{peg} & \small\text{cups} \newline \hline
> \hline
> \small\textbf{PerAct} & \small36 & \small24 & \small44 & \small12 & \small16 & \small20 & \small48 & \small0 & \small0 \newline
> \small\textbf{RVT} & \small29 & \small48 & \small91 & \small91 & \small50 & \small\textbf{36} & \small\textbf{100} & \small11 & \small26 \newline
> \small\textbf{Act3D} & \small12 & \small47 & \small95 & \small80 & \small51 & \small8 & \small99 & \small27 & \small9 \newline \hline
> \small\textbf{2D Diffuser Actor} & \small1 & \small48 & \small57 & \small68 & \small3 & \small0 & \small9 & \small6 & \small0 \newline
> \small\textbf{3D Diffuser Actor without Rel. Attn.} & \small35 & \small48 & \small97 & \small94 & \small64 & \small7 & \small98 & \small6 & \small12 \newline
> \small\textbf{3D Diffuser Actor (ours)} & \small\textbf{68} & \small\textbf{91} & \small\textbf{99} & \small90 & \small\textbf{74} & \small1 & \small99 & \small\textbf{79} & \small\textbf{34} \newline
> \end{array}

---

> ### Author Response · Authors · 2023-11-23
> **General Response (7)**
>
> ### Multi-task real-world experiments, baselines for the real-world experiments
>
>
> Thank you for your feedback. To satisfy your concern we conducted multi-task real-world experiments and we  compare our 3D Diffuser Actor with Act3D under such a multi-task setup.  We consider the folowing five tasks: 1. pick up bowls, 2. put grapes in the bowl, 3. flip bottle, 4. fold towel, 5. open oven. We use 20 demonstations per each task to train, and 10 for testing. We show results in the Table below, which we will add in a section in our final version.
>
>
> \begin{array}{l|cccc}
>  & \text{Flip bottle} & \text{Pick-up bowls} & \text{Put grapes} & \text{Fold towel} & \text{Open oven} \newline
> \hline
> \text{3D Diffuser Actor} & 100 & 100  & 50 & 60 & 20 \newline
> \text{Act3D} & 100 & 70 & 60 & 50 & 10 \newline
> \end{array}
>
> We see 3D Diffuser Actor outperforms Act3D also in the real world. This is not surprising as real world demonstations are even more multimodal than the ones in the RLbench simulator. We are working on collecting more real world tasks for our model, to match the task diversity of Per-Act's real world setup.
>
> ---
>
> ### Comparison with GNFactor
>
> Reviewer sG3K asked us to compare with GNFactor (Ze et al. CoRL 2023).
> The contribution of GNFactor is to train a neural network to complete a 3D feature volume given a single view. Given the complete 3D feature volume, they use an action prediction head identical to Per-Act. Thus, **GNFactor's contribution is orthogonal and complementary  to 3D Diffuser Actor**: we can use such 3D feature cross-view completion with our own 3D Diffuser Actor action prediction head and get similar performance boosts.  Thus comparing an action prediction head with a 3D feature completion method may not be an apples-to-apples comparison.
> However, we believe it is very valuable to still compare between the two, to also  satisfy the reviewer's concern. To do that,  we are training a version of 3D Diffuser Actor that uses:
> * a single camera (instead of 4 cameras)
> * 10 task selected by GNFactor (instead of 18)
> * 20 demos per task (instead of 100)
>
> The Table below shows our manipulation results, we also include GNFactors for comparison. We believe the  setup is identical to the one GNFactor considers.
>
>
> \begin{array}{l|ccccccccccc}
>  & \small\text{Avg.} & \small\text{close} & \small\text{open} & \small\text{sweep to} & \small\text{meat off} & \small\text{turn} & \small\text{slide} & \small\text{put in} & \small\text{drag} & \small\text{push} & \small\text{stack} \newline
>  & \small\text{Success.} & \small\text{jar} & \small\text{drawer} & \small\text{dustpan} & \small\text{grill} & \small\text{tap} & \small\text{block} & \small\text{drawer} & \small\text{stick} & \small\text{buttons} & \small\text{blocks} \newline
> \hline
> \text{3D Diffuser Actor} & \small 72 & \small 46 & \small 90  & \small 100 & \small 75 & \small 69 & \small 86 & \small 88 & \small 99 & \small 71 & \small 0 \newline
> \text{GNFactor} & \small 31.7 & \small 25.3 & \small 76.0 & \small 28.0 & \small 57.3 & \small 50.7 & \small 20.0 & \small 0.0 & \small 37.3 & \small 18.7 & \small 4.0 \newline
> \end{array}
>
> As you can see, **3D diffuser Actor without any 3D feature completion dramatically outperforms  GNFactor** that uses a Per-Act action prediction head. The results show that 3D Diffuser Actor is robust to the number of views used. Indeed, in the real world, 3D Diffuser Actor  considers a single viewpoint. We would like to thank Reviewer sG3K for urging us in this comparison. We will  include these results in our manuscript.

---

> ### Author Response · Authors · 2023-11-23
> **General Response (8)**
>
> ### RWs6y: explicitly demonstrate the multi-modality robot behavior qualitatively or quantitatively
> We believe the curves comparing 3D Diffuser with regression show quantitatively that diffusion helps over regression. For qualitative results you can see our Figure 1 and our supplementary videos. In Figure 1, you can see that the model has learnt 3 different modes for the same task, two of which grasp the same object in two different ways and one that grasps a different object first. Our method predicts keyposes, not continuous trajectories for the robot's end-effector, that is why you see not continuous trajectories visualized in the figure, but rather, next possible keyframes.
>
> ---
>
> ### RWs6y: Incremental contribution, trivial extension to Act3D
> Reviewer Ws6y mentions: "However, as the scene representation it uses can be trivially adapted from Act3D, replacing the regression policy head in Act3D with a diffusion model is an incremental contribution."
> The rebuttal we submitted, as well as the additional experiments and ablations we showed above suggest this is not the case.
>
> ---
>
> ### Summary
> We have presented a model that uses 3D representations and rendering of the estimates of a 3D gripper in the scene in a 3D denoising transformer model for learning multi-task manipulation policies from demonstations. The model works well with varying number of input views. It has state of the art performamce with a wide margin against the previous sota (12%) in RLbench, and it also works very well in the real world. It is general and principled. We thank the reviewers for pushing us to improve our work by asking for clearer writing, additional experiments,  qualitative results and comparisons with latest methods.

---

### Meta-Review · Area_Chair_Y7Py · 2023-12-11

**Metareview:**

The work proposes to combine diffusion model and 3D representation for robot manipulation policy learning. The method shows advantage by incorporating a 3D representation on the RLBench benchmark. However, there are a few concerns raised by reviewers on the significance of novelty, clarity, missing important details, and other aspects. The authors made a strong rebuttal, but the reviewers are still not convinced. The area chair also has additional concerns over the generalizability of the learned policy, which can be better evaluated. Based on the reviews and discussions, the area chair encourages the authors to resubmit the work by addressing the comments by reviewers.

**Justification For Why Not Higher Score:**

There are a number of unaddressed concerns by the reviewers.

**Justification For Why Not Lower Score:**

N/A

---

### Decision · Program_Chairs · 2024-01-16

Reject